# RiboPO: Preference Optimization for Structure- and Stability-Aware RNA Design

## Abstract

Designing RNA sequences that reliably adopt specified three-dimensional structures while maintaining thermodynamic stability remains challenging for synthetic biology and therapeutics. Current inverse folding approaches optimize for sequence recovery or single structural metrics, failing to simultaneously ensure global geometry, local accuracy, and ensemble stability—three interdependent requirements for functional RNA design. This gap becomes critical when designed sequences encounter dynamic biological environments. We introduce **RiboPO**, a **Ribo**nucleic acid **P**reference **O**ptimization framework that addresses this multi-objective challenge through *reinforcement learning from physical feedback* (RLPF). RiboPO fine-tunes gRNAde by *constructing preference pairs from composite physical criteria* that couple global 3D fidelity and thermodynamic stability. Preferences are formed using structural gates, *pLDDT* geometry assessments, and thermostability proxies with variability-aware margins, and the policy is updated with Direct Preference Optimization (DPO). On RNA inverse folding benchmarks, RiboPO demonstrates a superior balance of structural accuracy and stability. Compared to the best non-overlap baselines, our multi-round model improves Minimum Free Energy (MFE) by **12.3%** and increases secondary-structure self-consistency (EternaFold scMCC) by **20%**, while maintaining competitive 3D quality and high sequence diversity. In sampling efficiency, RiboPO achieves **11% higher pass@64** than the gRNAde base under the conjunction of multiple requirements. A multi-round variant with preference-pair reconstruction delivers additional gains on unseen RNA structures. These results establish RLPF as an effective paradigm for structure-accurate and ensemble-robust RNA design, providing a foundation for extending to complex biological objectives.

## 1 Introduction

The inverse folding problem in RNA design addresses a fundamental computational challenge: determining which nucleotide sequence will reliably fold into a specified three-dimensional backbone structure (Tang et al., 2024; Tan et al., 2025). This problem represents the cornerstone of rational RNA engineering (Wong et al., 2024), enabling systematic design of functional molecules for gene regulation (Green et al., 2014), therapeutics (Yip et al., 2024), and synthetic biology applications (Pfeifer et al., 2023).

*Designability*—defined as the total number or density of sequences that fold into a specific target structure—is a critical property for functional RNA. Highly designable structures are easier to discover, more mutation-robust (England et al., 2003), and tend to inhabit thermodynamically favorable *designable basins* (regions in sequence space with a high density of valid solutions) (Li et al., 1996). Optimizing geometry alone may reach thin, fragile ridges in sequence space; coupling geometry with ensemble stability biases search toward these robust basins (Schuster et al., 1994).

Contemporary geometric neural networks have achieved significant progress on this task (Wu et al., 2023; Wu & Li, 2023; Liu et al., 2025a; Patil et al., 2024). Models such as gRNAde (Joshi et al., 2025) utilize graph neural networks with geometric vector perceptrons to capture spatial relationships within RNA tertiary structures, delivering impressive sequence recovery performance across benchmark datasets. However, unlike proteins that maintain relatively stable native folds, RNA

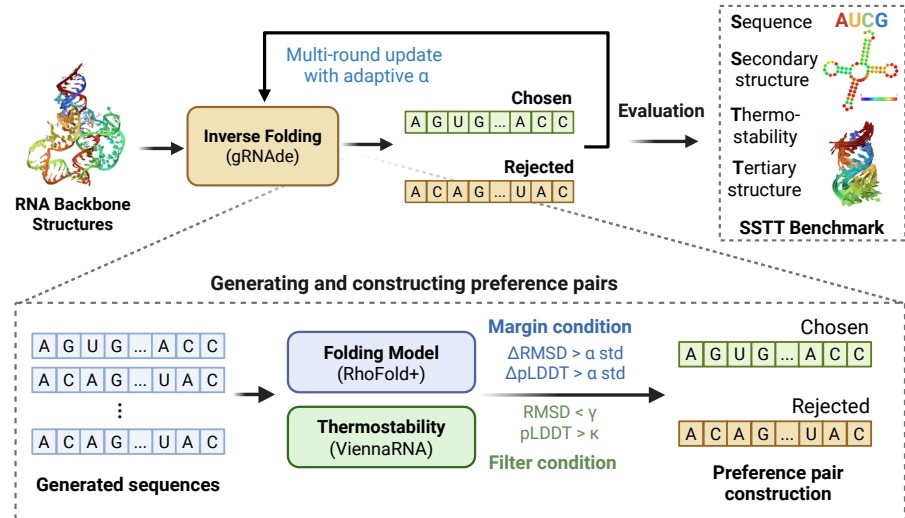

Figure 1: Overview of RiboPO framework.

molecules exhibit inherent conformational flexibility that causes them to sample multiple competing structures in solution (Ganser et al., 2019; Zadeh et al., 2011).

Current inverse folding methods (Hou et al., 2025; Wong et al., 2024), when optimized for sequence similarity or a single structural proxy, often neglect *designability* and thermodynamic stability, and therefore *do not* consistently adopt the intended conformation under physiological conditions.

Prevailing evaluations emphasize geometric correspondence (e.g., TMscore (Zhang & Skolnick, 2004) and RMSD for global alignment, pLDDT (Jumper et al., 2021) for local fidelity), but these alone provide limited insight into *ensemble viability* and *designability*. As a result, sequences can score well structurally yet occupy narrow, unstable basins that admit competing conformations in solution (Bernard et al., 2024). This evaluation-centric limitation produces sequences that achieve excellent structural scores while exhibiting poor ensemble behavior, sampling multiple competing conformations, and failing to demonstrate practical utility in biological applications (Zhou et al., 2023). From a design perspective, single-objective optimization cannot fully address the multi-faceted challenges inherent to RNA structural stability, where geometric fidelity must be balanced against thermodynamic preferences and conformational robustness.

To address these limitations, we present **RiboPO**, a reinforcement-learning–from–physical-feedback framework for RNA tertiary inverse folding that *explicitly* incorporates thermodynamic stability alongside geometry. Our approach reframes RNA inverse folding as a multi-objective preference optimization problem, representing a paradigm shift from purely geometric to biophysically-informed design. Unlike previous RL applications limited to RNA secondary structure prediction, RiboPO operates on full tertiary structures while balancing structural accuracy with thermodynamic stability. Our contributions are threefold:

- We incorporate thermodynamic stability measures into the reward for RNA inverse folding, representing the first systematic approach to include ensemble stability properties that existing RNA design methods have consistently overlooked. Our framework employs offline RL with precomputed training pairs, enabling efficient use of costly biophysical calculations during training.

- Second, our multi-objective preference optimization framework enables iterative refinement across competing design objectives through successive training rounds. This approach allows dynamic balancing between structural fidelity, local interaction accuracy, and thermodynamic stability without requiring online structure prediction during training.

- Third, we establish the **SSTT Benchmark** (**S**equence, **S**econdary, **T**ertiary, **T**hermostability) for comprehensive evaluation of RNA sequence generation quality. Our analysis reveals fundamental limitations in current geometric-only approaches and demonstrates that preference-based opti-

mization significantly improves both designability and thermodynamic stability while maintaining competitive sequence recovery performance.

On the DAS split (Das et al., 2010; Joshi et al., 2025) with our comprehensive SSTT suite, RiboPO achieves *general* improvements across dimensions: secondary-structure self-consistency (EternaFold scMCC: $+0.12$, **+20%**; 0.72 vs. 0.60) and thermostability (MFE: $-4.03$ kcal/mol, **12.3%**; $-36.86$ vs. $-32.83$), while maintaining competitive recovery and strong tertiary fidelity (RMSD **10.23**Å vs. 10.66Å; %RMSD$\leq 8$Å: **0.51** vs. 0.46). In Appendix A.5 (Pareto Analysis), we show that these shifts correspond to movement into more *designable basins* (lower energy, higher structural consistency). In §4 (Ablations), we validate the necessity of (i) variability-aware pair construction, (ii) DPO+$\lambda_{\text{SFT}}$ loss, and (iii) the multi-round schedule. Practically, sampling efficiency improves substantially.

Overall, preference-based optimization with physical feedback yields sequences that are not only structurally accurate but also ensemble-robust—advancing RNA design toward practically viable, mutation-tolerant solutions.

## 2 RELATED WORK

**RNA structure prediction and inverse folding.** Recent models have advanced RNA structure prediction across tertiary and secondary levels. For tertiary structure, traditional methods like DR-Fold (Li et al., 2023) and DeepFoldRNA (Pearce et al., 2022) combine primary/secondary structure information with energy functions for prediction and optimization. AlphaFold2-inspired approaches including NuFold (Kagaya et al., 2025), trRoseTTARNA (Wang et al., 2023) and Rho-Fold+ (Shen et al., 2024) employ EvoFormer architectures to predict structures from sequence and secondary structure data (Jumper et al., 2021). RoseTTAFoldNA (Baek et al., 2024) and AlphaFold 3 (Abramson et al., 2024) extend prediction capabilities to protein-nucleic acid complexes and multi-component assemblies. For secondary structure, traditional methods RNAfold/ViennaRNA (Lorenz et al., 2011) calculate partition functions and base-pairing probabilities through minimum free energy optimization. SPOT-RNA (Singh et al., 2019) uses 2D CNNs with transfer learning for non-canonical base pairs, E2EFold (Chen et al., 2020) predicts contact matrices, and RFold (Tan et al., 2024a) employs bidirectional decomposition to constrain cross-pairing interactions. Current inverse folding methods predominantly use GNN-based structural encoders, including RNAinformer (Patil et al., 2024), R3Design (Tan et al., 2025), RDesign (Tan et al., 2024b), RhoDesign (Wong et al., 2024), RiFold (Liu et al., 2025a), AlignIF (Wang et al., 2025a), and gRNAde (Joshi et al., 2025). RiboDiffusion (Huang et al., 2024) alternatively employs structure-conditioned discrete diffusion models. Notable advances include RhoDesign's (Wong et al., 2024) experimental validation, Ri-Fold's language modeling framework (Liu et al., 2025a), and gRNAde's (Joshi et al., 2025) extension from single-state to multi-state design.

**Reinforcement Learning and Preference Optimization for Inverse Folding.** RL and preference optimization are increasingly used to steer inverse folding toward structural fidelity and stability. ProteinZero explores online RL for self-improvement with proxy rewards (Wang et al., 2025b), while ResiDPO introduces residue-level Direct Preference Optimization (DPO) using structure-based rewards to improve designability (Xue et al., 2025). Diversity-regularized DPO has been applied to peptide inverse folding (Park et al., 2024). PLM-RL combined RL with the Protein Language Models (Cao et al., 2025b). RL has also been combined with diffusion models: RL-DIF optimizes structure-conditioned categorical diffusion (Ektefaie et al., 2024), and DRAKES back-propagates rewards through discrete diffusion trajectories for DNA/protein design (Wang et al., 2024). Search-based strategies like ProtInvTree use reward-guided tree search (Liu et al., 2025b), and lightweight, gradient-free finetuning for discrete sequences is explored in GLID$^2$E (Cao et al., 2025a). Finally, EnerBridge-DPO integrates energy-based preferences via Markov bridges with DPO to bias designs toward low-energy sequences (Rong et al., 2025). To the best of our knowledge, none of existing methods has demonstrated the effectiveness of RLPF on RNA.

## 3 METHODS

### 3.1 PROBLEM FORMULATION AND BACKBONE-CONDITIONED APPROACH

Ribonucleic acid (RNA) is a nucleic acid present in all living cells. An RNA molecule, often single-stranded, has a backbone made of alternating phosphate groups and sugar ribose. RNA has four nucleotides (also known as bases): adenine (A), guanine (G), cytosine (C), and uracil (U).

We formulate RNA inverse folding as a multi-objective optimization problem over the discrete RNA sequence space $\mathcal{S} = \{A, U, C, G\}^L$ ($L$ is the RNA sequence length), where we seek sequences that jointly maximize structural fidelity and thermodynamic stability:

$$s^* = \arg \max_{s \in \mathcal{S}} \left[ f_{\text{struct}}(s, \mathcal{G}) + f_{\text{thermo}}(s) \right], \tag{1}$$

where $\mathcal{G}$ denotes the target backbone geometry, $f_{\text{struct}}$ captures structural quality metrics (e.g., pLDDT, RMSD), and $f_{\text{thermo}}$ quantifies thermodynamic favorability via free energy.

To solve this, we use a backbone-conditioned policy that generates sequences given $\mathcal{G}$. Specifically, our policy $\pi_\theta(s \mid \mathcal{G})$ follows the GNN-GVP architecture from gRNAde (Joshi et al., 2025), encoding the 3D backbone and producing sequences autoregressively:

$$\pi_\theta(s \mid \mathcal{G}) = \prod_{i=1}^{L} \pi_\theta(s_i \mid s_{<i}, \mathcal{G}). \tag{2}$$

We maintain a frozen reference policy $\pi_{\text{ref}}$ initialized from the pre-trained gRNAde model to provide a stable baseline and regularize distributional drift.

Optimization proceeds via multi-round Direct Preference Optimization, which progressively shifts $\pi_\theta$ toward *more optimized solutions*. After each round, the improved policy supplies higher-quality preference pairs for the next round, yielding an iterative curriculum that stabilizes training while aligning the generation distribution with the multi-objective targets above.

### 3.2 MULTI-ROUND PREFERENCE CONSTRUCTION

**Progressive Refinement Strategy.** Our iterative approach leverages improved policies from previous rounds to generate candidate sequences in previously unexplored regions of sequence space (Belanger et al., 2019). This addresses the exploration-exploitation trade-off inherent in RNA sequence optimization, where complex structure-function relationships create numerous local optima (Hofacker et al., 2010; Jiménez et al., 2013). Single-round optimization frequently converges to sequences with adequate structural similarity but suboptimal thermodynamic properties or limited designability under physiological conditions. By progressively refining discrimination criteria across rounds, our method identifies increasingly subtle quality differences that drive convergence toward globally optimal solutions, balancing structural accuracy with biophysical stability (González et al., 2017). We operationalize this progressive refinement through round-wise tightening of preference criteria, as detailed below.

**Preference Criteria.** We establish high-quality preference pairs using systematic statistical thresholds tailored to the RNA inverse folding task. Given two sequences $s^w$ (winner/chosen) and $s^l$ (loser/rejected), a preference is recorded only when both filter and margin conditions are satisfied:

$$\textbf{Filter:} \quad \text{pLDDT}(g(s^w)) > 0.70 \wedge \text{RMSD}(\mathcal{G}, g(s^w)) < 8\,\text{Å} \tag{3}$$

$$\textbf{Margin:} \quad \underbrace{\left( \bigwedge_{m \in \{\text{pLDDT, RMSD}\}} |m(s^w) - m(s^l)| > \gamma_r \cdot \sigma_m \right)}_{\text{Structural Margin}} \wedge \underbrace{\left( \text{MFE}(s^w) < \text{MFE}(s^l) \right)}_{\text{Thermodynamic Preference}} \tag{4}$$

where $g$ denotes RNA folding approaches (RhoFold+ (Shen et al., 2024) in this work). $\sigma_m$ is the standard deviation of metric $m(\cdot)$ estimated from the current candidate pool. These criteria exclude

candidates unlikely to fold correctly, ensuring that preference labels are assigned within a biologically realistic design space and reducing downstream noise. The statistical margin $\gamma_r \sigma_m$ enforces the *significance* of the structural preferences, effectively improving the signal-to-noise ratio of the implicit reward; normalization by $\sigma_m$ renders metrics on different scales commensurate and stabilizes the DPO likelihood-ratio updates.

Concretely, we implement this progressive refinement of structure and energy by decreasing $\gamma_r$ across iterations. Early rounds use looser margins to rapidly establish broad foldability; later rounds tighten the margins to capture subtle, near–native improvements (e.g., local geometry). This round-wise annealing functions as Curriculum Learning (CL): large early gaps yield high-confidence preferences that reduce label noise and reward sparsity, while smaller later gaps provide denser, finer-grained gradient signals. The schedule mirrors an exploration-to-exploitation transition and implicitly enforces a trust-region constraint relative to the reference policy, stabilizing updates of large autoregressive models.

### 3.3 REINFORCEMENT LEARNING FROM PHYSICAL FEEDBACK VIA DPO

We construct preference pairs $(s^w, s^l)$ by sampling candidate sequences from the current policy $\pi_\theta(\cdot \mid \mathcal{G})$ for each target RNA backbone $\mathcal{G}$ and then applying the filter and margin criteria described above. This procedure ensures that $s^w$ and $s^l$ reflect biologically realistic, thermodynamically feasible regions of sequence space.

Our objective maximizes the preference likelihood ratio between $s^w$ and $s^l$ while anchoring against the reference policy:

$$\mathcal{L}_{\mathrm{DPO}} = -\mathbb{E}\left[\log \sigma\left(\beta \log \frac{\pi_\theta(s^w \mid \mathcal{G})}{\pi_\theta(s^l \mid \mathcal{G})} - \beta \log \frac{\pi_{\mathrm{ref}}(s^w \mid \mathcal{G})}{\pi_{\mathrm{ref}}(s^l \mid \mathcal{G})}\right)\right]. \tag{5}$$

where it directly shapes the policy toward generating sequences with better structural fidelity and thermodynamic stability. while the reference-policy term acts as a trust region to prevent destabilizing updates, preventing regressions toward suboptimal folds.

To stabilize training further, we add supervised fine-tuning (SFT) loss on the preferred sequences:

$$\mathcal{L} = \mathcal{L}_{\mathrm{DPO}} + \lambda_{\mathrm{SFT}} \, \mathbb{E}_{s^w}\left[-\log \pi_\theta(s^w \mid \mathcal{G})\right], \tag{6}$$

where $\lambda_{\mathrm{SFT}}$ is a hyperparameter controlling the strength of this term. The SFT component acts as an explicit maximum-likelihood "anchor" on sequences with higher designability and thermostability potential, reducing catastrophic drift and mode collapse often seen in RL-based fine-tuning.

### 3.4 SSTT BENCHMARK: COMPREHENSIVE EVALUATION FRAMEWORK

Unlike existing RNA inverse folding benchmarks, which mainly evaluate sequence recovery and RMSD, we introduce the SSTT benchmark to assess performance across four complementary dimensions. This framework integrates thermodynamic stability and ensemble properties alongside traditional structural metrics, offering a multidimensional view of design quality. Detailed calculation procedures are provided in Appendix A.2. **S**equence Axis captures how closely designed sequences resemble native ones and how broadly they explore sequence space. We assess recovery rate (Dauparas et al., 2022) to measure native residue similarity, and 3-mer diversity (Shannon, 1948; Bokulich, 2024) to quantify sequence exploration capacity. **S**econdary Structure Axis evaluates whether designed sequences fold back into their intended secondary structures. We measure forward-folding self-consistency using Matthews Correlation Coefficient (Matthews, 1975; Chicco et al., 2021) on EternaFold (Wayment-Steele et al., 2022) predictions, capturing base-pairing fidelity. **T**ertiary Structure Axis assesses three-dimensional fidelity at both global and local levels as well as steric realism. We evaluate TM-score (Zhang & Skolnick, 2004), GDT, and RMSD for global similarity; pLDDT (Jumper et al., 2021) for confidence; interaction network fidelity (Parisien et al., 2009) for base-pairing accuracy; and clash score (Word et al., 1999; Davis et al., 2007) for geometric feasibility. **T**hermostability Axis quantifies the thermodynamic robustness of designed sequences under physiological conditions. We incorporate minimum free energy (Zuker & Stiegler, 1981) for stability, and ensemble defect (Zadeh et al., 2011) to measure target structure specificity.

## 4 EXPERIMENTS & RESULTS

### 4.1 EXPERIMENTAL SETTINGS

**Data Processing** Our experimental setup aligns with established DAS benchmark splits (Das et al., 2010) for compatibility with existing evaluation protocols (Joshi et al., 2025). We implement strict length consistency filtering to remove preference pairs where sequence lengths differ or do not match graph node counts, ensuring training stability and preventing spurious length-based correlations. Following preprocessing, our dataset comprises 20,811 training pairs, 505 validation pairs, and 429 test pairs, drawn from 4,025 training instances, 100 validation instances, and 98 test instances. This testing dataset split has overlap with the training dataset of RiboDiffusion (Huang et al., 2024) and RDesign (Tan et al., 2024b), which leads to the abnormally high sequence recovery (100%) for some structures like 7D7W.

**Evaluation Protocol and Metrics** We evaluate all approaches, generating 8 sequence candidates under temperature 0.1 per backbone for statistical reliability. We directly apply the SSTT evaluation pipeline (Section 3.4) with EternaFold (Wayment-Steele et al., 2022) for secondary structure predictions, and RhoFold+ (Shen et al., 2024) for tertiary structure prediction. Furthermore, we report thresholded success rates, including percentages achieving RMSD $\leq$ 15.0Å, pLDDT $\geq$ 0.50, MFE $\leq$ -8.0 $kcal/mol$, and TM $\geq$ 0.05.

We evaluated pass@k performance (Lyu et al., 2024; Wu et al., 2025) by sampling 64 sequences per RNA structure (folded by RhoFold+ (Shen et al., 2024)) and measuring success rates across different k values (1, 2, 4, 8, 16, 32, 64). For a given structure and k value, pass@k is defined as:

$$\text{pass@k} \; = \; 1 - (1 - p)^k. \tag{7}$$

where p denotes the indicator function, returning 1 when the case pass the criteria, and returning 0 otherwise.

**Baseline Approaches** We compare RiboPO against recent deep learning based RNA inverse folding models spanning two main paradigms. **Autoregressive models** include gRNAde (Joshi et al., 2025), RDesign (Tan et al., 2024b), R3Design (Tan et al., 2025), RiFold (Liu et al., 2025a), and RhoDesign (Wong et al., 2024). **Diffusion-based models** include RiboDiffusion (Huang et al., 2024), RhoDesign (Wong et al., 2024) and **RIdiffusion** (Hou et al., 2025).

### 4.2 SSTT BENCHMARK ANALYSIS

**Baseline overview.** We evaluate all four SSTT dimensions comprehensively (Table 1). Among baselines, **gRNAde** delivers the strongest sequence- and secondary-structure metrics, reflecting its GNN–GVP backbone conditioning. **Single-round RiboPO improves secondary and thermodynamic properties.** Our first-round DPO (**RiboPO Round 1**) shows that even a single training round markedly enhances secondary-structure and thermodynamic properties over the gRNAde base model. Despite a minor decline in recovery (0.5288 → 0.504), RiboPO increases diversity (0.83 → **0.90**) and improves scMCC (0.60 → 0.71) and MFE (−32.83 → −35.83). This indicates that policy optimization effectively trades off marginal sequence similarity for substantially better foldability and energetic favorability. This represents early-stage exploration of a broader biophysically plausible sequence with improved stability. **Progressive refinement through multiple rounds.** As RiboPO proceeds to multiple rounds, **thermostability gains become striking**. By Round 2, MFE further improves to −36.86 **kcal/mol**, the best of all models, while scMCC remains high (0.70) and tertiary metrics (RMSD and %RMSD≤8Å) also improve. These gains demonstrate that decreasing $\gamma_r$ across rounds—our "coarse-to-fine" margin schedule—admits increasingly subtle yet energetically favorable sequences, allowing the model to internalize thermodynamic regularities and better capture RNA structure–stability trade-offs. **Balanced sequence and structural fidelity.** Although RiboPO prioritizes stability and secondary structure, its tertiary metrics remain competitive. Across all rounds, pLDDT holds steady at ∼0.65, RMSD stays near 10.4Å, and TM-score remains comparable to or better than several baselines. This suggests that thermodynamic optimization does not compromise 3D fidelity; instead, multi-round training preserves global structure while refining local and energetic features. In RNA terms, the model learns not only to produce sequences that fold correctly but also to maintain a realistic conformational ensemble.

Table 1: Main benchmark results on the DAS test set. RiboPO significantly outperforms baselines on thermostability and secondary structure metrics (T and S), while remaining competitive on sequence and tertiary structure metrics (S and T). Best results are in **bold**.

| Model | S (Sequence) | | S (2D Struct.) | T (3D Struct.) | | | | | T (Thermo.) |
|---|---|---|---|---|---|---|---|---|---|
| | Rec. ($\uparrow$) | Div. ($\uparrow$) | scMCC ($\uparrow$) | pLDDT ($\uparrow$) | %pLDDT $\geq 0.70$ | RMSD ($\downarrow$) | %RMSD $\leq 8$Å | TM-sc. ($\uparrow$) | MFE ($\downarrow$) |
| RiFold | 0.416 | 0.89 | 0.28 | 0.50 | 0.012 | 17.06 | 0.20 | 0.12 | -11.34 |
| RDesign | 0.415 | 0.84 | 0.20 | 0.45 | 0.22 | 16.81 | 0.12 | 0.12 | -10.59 |
| RIDiffusion | **0.533** | 0.85 | 0.59 | 0.60 | 0.28 | 10.66 | 0.46 | 0.25 | -28.81 |
| gRNAde (Base) | 0.529 | 0.83 | 0.60 | 0.62 | 0.38 | 11.45 | 0.42 | 0.29 | -32.83 |
| RiboPO (Round 1) | 0.504 | **0.90** | 0.71 | **0.66** | **0.45** | 10.40 | 0.45 | 0.30 | -35.83 |
| RiboPO (Round 2) | 0.502 | 0.88 | 0.70 | 0.65 | 0.45 | **10.23** | **0.51** | **0.31** | **-36.86** |
| RiboPO (Round 4) | 0.506 | 0.89 | **0.72** | 0.65 | 0.42 | 10.42 | 0.44 | 0.29 | -35.22 |

Table 2: Pass@k results for generating a sequence with RMSD < 12 Å, pLDDT > 0.50, TM-score > 0.05, and MFE < −4.0 kcal/mol (T = 0.5). RiboPO demonstrates superior sampling efficiency with Multi-round RiboPO achieving the best performance.

| Model | pass@1 | pass@2 | pass@4 | pass@8 | pass@16 | pass@32 | pass@64 |
|---|---|---|---|---|---|---|---|
| gRNAde | 0.308 | 0.346 | 0.385 | 0.577 | 0.654 | 0.731 | 0.731 |
| RiboPO (Round 1) | 0.346 | 0.462 | 0.538 | **0.692** | **0.731** | 0.769 | 0.769 |
| RiboPO (Round 4) | **0.423** | **0.500** | **0.538** | 0.654 | 0.654 | **0.808** | **0.808** |

## 4.3 ADVANCED ANALYSIS OF REINFORCEMENT LEARNING EFFECTS

**RiboPO improves hit rate and sampling efficiency.**     We assess sampling efficiency using pass@k analysis on the identified 14 important RNA structures by (Das et al., 2010) (Table 2). For small $k$ values (pass@1–4), RiboPO achieves 37%-44% relative increase in success rate. Even at large $k$ (pass@32–64) Figure 2a, RiboPO sustains 11% higher success rates. This demonstrates that DPO improves the "hit rate" of high-quality sequences per sampling budget, which is critical when each candidate must be computationally folded. From an RL perspective, this reflects curriculum-based refinement that concentrates probability mass in high-reward regions; from an RNA perspective, it means fewer trials are needed to find sequences satisfying structural and energetic constraints, thereby reducing in silico or wet-lab burden.

**Multi-round DPO drives measurable distribution shifts.**     Figure 2b compares gRNAde and RiboPO after multi-round training on TM-score, RMSD, pLDDT, and MFE. We observe that RMSD and MFE distributions move toward higher-quality regions: the incidence of poor outliers (RMSD > 15Å, MFE > −10 kcal/mol) decreases while the mass of stable, low-energy designs increases (mean MFE −16.5 → −18.9 kcal/mol). pLDDT, already relatively high for the base model (mean ∼ 0.62), becomes more stable with enrichment in the 0.65–0.8 range and suppression of low-confidence cases. TM-score exhibits a "center-convergence" pattern: scores at the low end are lifted, yet samples concentrate near the middle (0.15–0.40), consistent with indirect optimization via RMSD and pLDDT rather than TM-score itself. From the RL viewpoint, these shifts confirm that large early margins ($\gamma_r$ high) promote coarse exploration and smaller later margins refine toward explicitly rewarded regions (RMSD, MFE). From the RNA viewpoint, the policy first secures gross foldability and energetic feasibility, then favors finer conformational improvements; **this two-phase process mirrors natural RNA evolution**.

## 4.4 ABLATION STUDIES

**Ablation on loss components and hyperparameters.**     To test whether our loss formulation truly supports multi-objective optimization, we ablate both loss components and loss weights (Table 3). Removing the DPO term ($\beta$=0) causes broad degradation in structural metrics (pLDDT 0.62, RMSD 11.50), confirming that pairwise preference learning is the main driver of structural improvements. Ablating the SFT term ($\lambda_{SFT}$=0) increases diversity (0.93) and makes MFE more negative (−42.60 kcal/mol), but markedly worsens tertiary quality (TM-score 0.19, RMSD 13.55). This in-

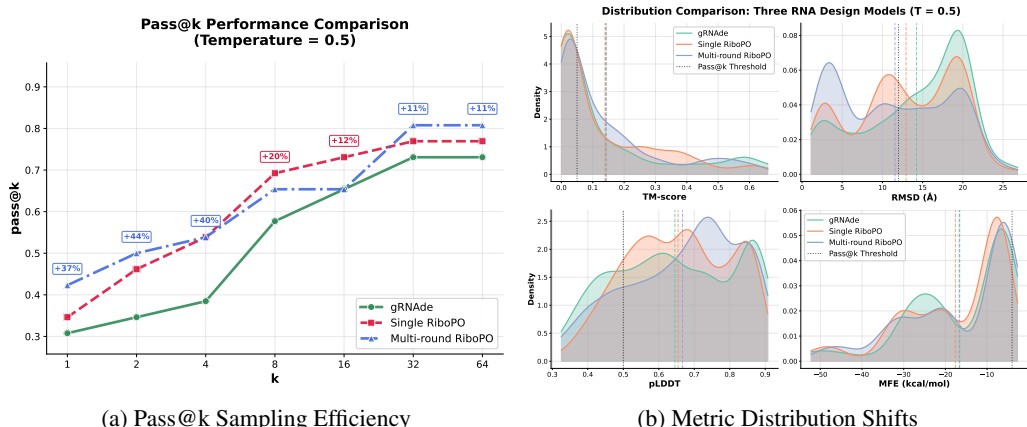

(a) Pass@k Sampling Efficiency      (b) Metric Distribution Shifts

Figure 2: **RiboPO improves both sampling efficiency and the quality distribution of generated sequences. (a)** Pass@k performance shows RiboPO consistently dominates gRNAde across all sample sizes $k$, with the largest gains at the small-$k$ screening regime crucial for practical applications. **(b)** Kernel density estimates for key metrics on 14 held-out RNA structures. Multi-round RiboPO shifts the distributions for RMSD and MFE towards more favorable values (indicated by vertical lines) compared to the gRNAde baseline, demonstrating improved designability and thermostability.

Table 3: Ablation study on loss components for single round training

| Configuration | S (Sequence) | | S (2D Struct.) | T (3D Struct.) | | | | | T (Thermo.) |
|---|---|---|---|---|---|---|---|---|---|
| | Rec. ($\uparrow$) | Div. ($\uparrow$) | scMCC ($\uparrow$) | pLDDT ($\uparrow$) | %pLDDT $\geq 0.70$ | RMSD ($\downarrow$) | %RMSD $\leq 8$ Å | TM-sc. ($\uparrow$) | MFE ($\downarrow$) |
| w/o $\mathcal{L}_{\mathrm{DPO}}$ ($\beta = 0$) | 0.49 | 0.85 | 0.61 | 0.62 | 0.36 | 11.50 | 0.36 | 0.27 | -33.68 |
| w/o $\mathcal{L}_{\mathrm{SFT}}$ ($\lambda_{SFT} = 0$) | 0.44 | 0.93 | 0.64 | 0.62 | 0.42 | 13.55 | 0.39 | 0.19 | **-42.60** |
| RiboPO ($\beta = 0.5$) | 0.47 | 0.88 | 0.57 | 0.62 | 0.40 | 11.96 | 0.45 | 0.25 | -41.05 |
| RiboPO ($\beta = 1.0$) | 0.49 | 0.85 | 0.59 | 0.60 | 0.35 | 13.90 | 0.26 | 0.20 | -41.86 |
| RiboPO ($\beta = 0.12$) | **0.50** | **0.90** | **0.71** | **0.66** | **0.45** | **10.40** | **0.45** | **0.30** | -35.83 |

dicates that SFT functions as a distributional anchor that prevents over-exploitation of low-energy but geometrically unrealistic regions; **removing SFT trades stability for 3D realism**.

Loss-weight ablation further shows a clear trade-off governed by $\beta$. According to the DPO objective, a larger $\beta$ weakens the KL regularization against the reference policy $\pi_{ref}$, allowing the model to deviate more aggressively toward aligning with the chosen responses. Conversely, smaller $\beta$ values enforce a stronger trust region around $\pi_{ref}$. The detailed theoretical analysis is at Appendix A.9. Consequently, the setting $\beta = 0.12$ achieves highly optimized performance across most evaluation metrics. By contrast, $\beta = 0.5$ yields slightly better performance than $\beta = 1.0$, reflecting a more balanced optimization between fidelity to $\pi_{ref}$ and adaptation to preferences. In the case of MFE, the performance exhibits an inverse relationship with $\beta$: larger values consistently result in superior outcomes. We attribute this to the construction of the pairwise preference data, which introduces no explicit margin for MFE, thereby weakening the available learning signal. In contrast, pLDDT and RMSD provide stronger and more discriminative gradients that dominate optimization when $\beta$ is small, leading to poorer MFE. At smaller $\beta$ values, the stronger KL regularization keeps the policy closer to $\pi_{ref}$ while preference updates are dominated by structural margins, which limits how much MFE can improve. As $\beta$ increases and the KL term weakens, the model can deviate further from $\pi_{ref}$ and better exploit whatever signal about thermodynamic stability is present in the preferences, so MFE benefits disproportionately even though other structural metrics may plateau or decline. These observations highlight that the distribution of the preference dataset plays a pivotal role in shaping DPO performance, which is consistent with recent findings in (Pan et al., 2025).

**Ablation on offline RL algorithms.** We compare SimPO and DPO under a single round (Table 4). DPO consistently outperforms SimPO on structural fidelity (scMCC 0.71 vs. 0.66, RMSD 10.40 vs. 11.36, TM-score 0.30 vs. 0.27) and on robustness thresholds (%RMSD≤8Å: 0.45 vs. 0.40;

Table 4: Comparison of different RL algorithms (DPO and SimPO) with only a single round. DPO demonstrates superior performance across key metrics.

| Algorithm | S (Sequence) | | S (2D Struct.) | T (3D Struct.) | | | | | T (Thermo.) |
|---|---|---|---|---|---|---|---|---|---|
| | Rec. (↑) | Div. (↑) | scMCC (↑) | pLDDT (↑) | %pLDDT ≥0.70 | RMSD (↓) | %RMSD ≤8Å | TM-sc. (↑) | MFE (↓) |
| SimPO | **0.51** | 0.87 | 0.66 | 0.65 | 0.44 | 11.36 | 0.40 | 0.27 | **-36.28** |
| DPO | 0.50 | **0.90** | **0.71** | **0.66** | **0.45** | **10.40** | **0.45** | **0.30** | -35.83 |

Table 5: Ablation study on multi-round training components. Updating the reference model and using a curriculum for the preference margin are both beneficial. MR: Multi-Round; CL: Curriculum Learning, with narrowing preference margin; Take the checkpoint evaluation results at the end of final iteration.

| Strategy | S (Sequence) | | S (2D Struct.) | T (3D Struct.) | | | | | T (Thermo.) |
|---|---|---|---|---|---|---|---|---|---|
| | Rec. (↑) | Div. (↑) | scMCC (↑) | pLDDT (↑) | %pLDDT ≥0.70 | RMSD (↓) | %RMSD ≤8Å | TM-sc. (↑) | MFE (↓) |
| Multi-Round DPO, w/o CL | 0.49 | **0.91** | 0.69 | 0.64 | 0.39 | 13.39 | 0.39 | 0.24 | -35.10 |
| Reference Model Update | 0.47 | 0.85 | 0.63 | 0.60 | 0.35 | 14.30 | 0.17 | 0.16 | -48.07 |
| Multi-Round CL SimPO | 0.46 | 0.87 | 0.61 | 0.58 | 0.25 | 13.14 | 0.32 | 0.21 | **-48.22** |
| Multi-Round CL DPO | **0.51** | 0.89 | **0.72** | **0.65** | 0.42 | **10.42** | 0.44 | 0.29 | -35.22 |

%pLDDT$\geq 0.70$: $0.45$ vs. $0.44$), while maintaining higher diversity ($0.90$ vs. $0.87$). SimPO shows slightly higher recovery ($0.51$ vs. $0.50$) and more negative MFE ($-36.28$ vs. $-35.83$). These results suggest that DPO's likelihood-ratio objective with an explicit reference anchor is better at preserving global 3D realism while still exploring diverse sequences; **SimPO favors stability and exact-token matching slightly more, but at the expense of tertiary structure**.

**Analysis of multi-round training strategy.** Table 5 disentangles three design choices: curriculum on the preference margin (CL), reference-model updates across rounds, and algorithm family. Multi-round DPO without curriculum already improves over the base, but exhibits moderate RMSD ($13.39$) and limited high-confidence structures (%pLDDT$\geq 0.70$=$0.39$). Introducing a curriculum that narrows the margin across rounds (*Multi-Round CL DPO*) yields the strongest overall structural performance (scMCC $0.72$, pLDDT $0.65$, RMSD $10.42$, %RMSD$\leq 8$Å $0.44$, TM-score $0.29$) with good diversity ($0.89$). This aligns with our design: early coarse margins de-noise preferences; later fine margins admit subtle yet consistent improvements, steering the policy into near-native basins.

Updating the reference model each round dramatically pushes MFE downward ($-48.07$ kcal/mol) but harms structure (RMSD $14.30$, TM-score $0.16$). The moving reference weakens the trust-region effect and effectively "ratchets" the policy toward low-energy pockets even when geometry degrades, indicating reward hacking on stability proxies. A curriculum with SimPO across rounds also achieves very low MFE yet lags on structure, reinforcing that the objective form and anchoring are critical. **The best balance arises from *fixed-reference* multi-round DPO with a margin curriculum**, which preserves 3D fidelity while improving stability.

# 5 CONCLUSION

We introduced **RiboPO**, a backbone-conditioned preference optimization framework that treats RNA inverse folding as a multi-objective problem over structural fidelity and thermodynamic stability. On SSTT, RiboPO improves thermostability and secondary-structure self-consistency while remaining competitive on tertiary metrics, and raises pass@k hit rates—evidence of more efficient sampling. Ablations show that DPO drives structural gains, SFT controls distributional drift and preserves geometry, and the margin curriculum enables coarse-to-fine refinement.

Our analysis also clarifies limits of offline preference alignment: energy-leaning signals shift the policy toward low-MFE basins (explaining recovery drops), and 2D thermodynamic proxies do not fully align with global 3D similarity under current RNA predictors. **Overall, preference-based RL reliably steers sequences toward biophysically plausible, designable RNAs while preserving 3D fidelity.**

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

# A APPENDIX

## A.1 IMPLEMENTATION DETAILS AND COMPUTATIONAL EFFICIENCY

**Training details.** Our training protocol employs cosine learning rate decay with a 1,000-step warmup, gradient accumulation across 4 steps, a batch size of 32 per preference pair, and precision bf16 training. Training proceeds for a minimum of 5 epochs, with validation performed every 50 steps, and model checkpoints saved at the same interval.

**Computational Efficiency Analysis.** We analyze the runtime costs for both inference (deployment) and training (development) to demonstrate the practicality of the framework.

- **Inference Latency:** We benchmarked inference speeds on a single **NVIDIA RTX A6000 GPU** using the 4FE5 backbone (Length=67). For a batch size of 8, RiboPO (based on gR-NAde) required 5.29 seconds ($\approx 0.66$ s/seq). This is comparable to the diffusion baseline RiboDiffusion (Huang et al., 2024), which required 2.68s with 50 denoising steps ($\approx 0.34$ s/seq) and 4.73s with 100 denoising steps ($\approx 0.59$ s/seq). While diffusion speed varies with step count ($T$), RiboPO's autoregressive generation provides deterministic runtime scaling with length ($L$), offering competitive throughput without the need to tune denoising hyperparameters.

- **Preference Pair Construction (Offline):** RiboPO utilizes an offline RL strategy, decoupling physical feedback from gradient updates. The construction of the preference dataset ($\sim 14,000$ pairs) is the primary computational bottleneck due to the cost of 3D structure prediction. This process required approximately 3 days using a 32-core Intel Xeon Gold 6530 CPU, utilizing RhoFold+ (Shen et al., 2024) for structure prediction.

- **Training Time and Convergence:** Once preference data is pre-computed, the DPO fine-tuning is efficient. A single round of training (10 epochs) requires approximately 16 hours on a single NVIDIA A100 GPU. We observed that the model converges rapidly, reaching stability in validation preference accuracy at approximately 8,000 gradient steps. This indicates high sample efficiency compared to online RL methods that often require millions of samples. For each sample, RhoFold+ predicts the 3D structure at around 0.14s (without MSA), and thus acting as an efficient proxy tool.

- **Sample Efficiency (The "Hit Rate" Payoff):** While RiboPO incurs an offline training cost, it significantly reduces the deployment cost. As shown in our Pass@k analysis, RiboPO achieves an 11% higher hit rate than the baseline. Consequently, a user needs to sample and fold fewer sequences to find a valid design, significantly reducing the "Total Time to Success" in experimental or in silico workflows.

**Asymptotic Time Complexity.** Let $L$ denote backbone length, $K$ the number of backbone conformations, and $T$ the number of denoising steps in a diffusion sampler. Both gRNAde and RiboPO use a $k$-nearest-neighbour geometric graph with $k = O(1)$ and a fixed number of GNN layers, so $|E| = O(L)$ and a single encoder–decoder forward pass scales as $O(KL)$. Generating $M$ sequences with RiboPO or gRNAde therefore costs $O(MKL)$, and DPO fine-tuning over $N_{\text{pair}}$ preference pairs has total cost $O(N_{\text{pair}}KL)$, up to a constant factor for the number of epochs. In contrast, RiboDiffusion (Huang et al., 2024) applies its backbone network for $T$ denoising steps, so sampling a single sequence scales as $O(TL)$, introducing an explicit multiplicative factor $T$ compared to the single-pass autoregressive cost. Finally, offline preference construction is dominated by external structure and energy predictions: if $C_{\text{3D}}(L)$ and $C_{\text{2D/thermo}}(L)$ denote the costs of 3D and thermodynamic engines for length $L$, constructing a dataset with $N_G$ backbones and $N_{\text{cand}}$ candidates per backbone has cost $O(N_G N_{\text{cand}}(C_{\text{3D}}(L) + C_{\text{2D/thermo}}(L)))$, which is a one-time offline expense reused across all DPO training runs.

## A.2 SSTT: COMPREHENSIVE FINE-GRAINED RNA DESIGN

**Notation.** For a candidate sequence $y = (y_1, \ldots, y_L)$ and the native sequence $y^{\text{nat}}$, let $S_0$ denote the target secondary structure. Partition-function outputs are base-pair probabilities $p_{ij}$ and unpaired probabilities $q_i$. Predicted 3D coordinates are $\mathbf{X}(y)$; native/target coordinates are $\mathbf{X}^{\text{ref}}$.

SEQUENCE AXIS (S)

- **Recovery** (Dauparas et al., 2022): Recovery measures the average percentage of native nucleotides correctly recovered in the sampled sequences. It is computed as

$$\text{Rec}(y) \;=\; \frac{1}{L}\sum_{i=1}^{L}\mathbb{1}\{y_i = y_i^{\text{nat}}\}.$$

- **Perplexity** ($\downarrow$, lower-is-better) (Bengio et al., 2003): Perplexity measures how well a model predicts the sequences. Given a model (reference or policy) with token probabilities $p_\theta$, perplexity is defined as the average exponential of the negative log-likelihood of the sampled sequences.

$$\text{PPL}(y) \;=\; \exp\Big(-\frac{1}{L}\sum_{i=1}^{L}\log p_\theta(y_i \mid y_{<i}, \mathcal{G})\Big).$$

- **Diversity (3-mer)** ($\uparrow$): Given $n$ candidates for the backbone $\mathcal{G}$, let $v_s \in \mathbb{R}^{64}$ be the normalized 3-mer frequency of candidate $s$. Define

$$\text{Div}_{\text{3mer}}(\mathcal{G}) \;=\; 1 \;-\; \frac{2}{n(n-1)}\sum_{1\le s<t\le n}\rho\big(v_s, v_t\big),$$

  where $\rho(\cdot,\cdot)$ is the Pearson correlation. It's the quantification of sequence variety based on substring frequency distributions (Shannon, 1948; Bokulich, 2024).

SECONDARY AXIS (S) — FORWARD-FOLDING SELF-CONSISTENCY

- **2D (EternaFold) scMCC (Matthews Correlation Coefficient) (Matthews, 1975)** ($\uparrow$): This metric measures the ability of designs to recover base pairing patterns. We use EternaFold (Wayment-Steele et al., 2022) to perform forward folding on the sampled sequences and then calculate the Matthews correlation coefficient (MCC) by comparing the predicted base-pair adjacency matrix with the ground truth.

THERMOSTABILITY AXIS (T) — ENSEMBLE SPECIFICITY

- **MFE** (kcal/mol, $\downarrow$): The minimum free energy (MFE) of the most stable predicted secondary structure. *Note:* MFE is a single-structure quantity and does not capture ensemble specificity; we therefore also report Ensemble defect per nucleotide (ED/nt).

- **Ensemble defect (ED) (Zadeh et al., 2011) and ED/nt** ($\downarrow$): The ensemble defect (ED) measures the discrepancy between the predicted base-pair probabilities and the true structural states.

$$\text{ED}(y; S_0) = \sum_{i \in U_0}(1 - q_i) + \sum_{i \in P_0}\big(1 - p_{i\,j^*(i)}\big), \qquad \text{ED/nt} = \text{ED}/L.$$

TERTIARY AXIS (T)

- **TM-score** (Zhang & Skolnick, 2004) ($\uparrow$): Evaluates structural alignment quality using US-align (Zhang et al., 2022).

$$\text{TM}(y) \;=\; \frac{1}{L_{\text{ref}}}\sum_{i=1}^{L_{\text{ali}}}\frac{1}{1 + \Big(\frac{d_i}{d_0(L_{\text{ref}})}\Big)^2}, \quad d_0(L) \;=\; 1.24\sqrt[3]{L - 15} - 1.8.$$

- **pLDDT** ($\uparrow$): The mean predicted LDDT confidence (Jumper et al., 2021).

- **RMSD** ($\downarrow$): Root-Mean-Square Deviation of the atomic coordinates.

- **INF (ALL)** (Parisien et al., 2009) ($\uparrow$): Interaction-network fidelity assessing base-pairing accuracy.

- **Clash score** ($\downarrow$): Number of all-atom steric clashes per 1000 atoms (Word et al., 1999).

## A.3 Full RiboPO SSTT Results and Baselines (Extended)

Table 6 presents the extended set of SSTT metrics not reported in the main text. This includes sequence diversity, global structure quality (GDT), thresholded success rates, confidence scores (pLDDT), interaction fidelity (INF), stereochemical feasibility (Clash Score), and ensemble specificity (ED/nt).

Table 6: Extended SSTT Benchmark Results. This table complements Main Table 1 by reporting additional metrics for Diversity, Tertiary Quality (GDT, pLDDT, INF), Stereochemistry (Clash Score), and Ensemble Specificity (ED/nt). RiboPO (Round 2) demonstrates the best balance of diversity (0.87) and structural confidence (pLDDT 0.65), while maintaining low steric clashes and competitive ensemble defects.

| Model | Div (3-mer) ↑ | GDT ↑ | %GDT ≥ 0.5 ↑ | %TM ≥ 0.45 ↑ | pLDDT ↑ | %pLDDT ≥ 0.7 ↑ | INF (ALL) ↑ | Clash ↓ | ED/nt ↓ |
|---|---|---|---|---|---|---|---|---|---|
| RiFold | 0.89 | 0.121 | 0.012 | 0.024 | 0.502 | 0.012 | 0.39 | 678.3 | 0.0034 |
| RDesign | 0.84 | 0.124 | 0.024 | 0.012 | 0.453 | 0.024 | 0.38 | 740.6 | 0.0036 |
| RIDiffusion | 0.85 | 0.272 | 0.157 | 0.168 | 0.609 | 0.282 | 0.52 | 644.0 | 0.0626 |
| gRNAde (Base) | 0.82 | 0.285 | 0.286 | 0.301 | 0.621 | 0.381 | 0.48 | 637.5 | 0.0028 |
| RiboPO Round 1 | **0.90** | 0.291 | 0.273 | 0.302 | **0.655** | **0.450** | **0.50** | **595.5** | **0.0021** |
| RiboPO Round 2 | 0.87 | **0.314** | **0.301** | **0.325** | 0.652 | 0.449 | 0.49 | 617.9 | 0.0043 |
| RiboPO Round 4 | 0.89 | 0.280 | 0.239 | 0.287 | 0.646 | 0.424 | 0.49 | 613.6 | 0.0043 |

## A.4 Evaluation on De-duplicated Benchmark

To ensure a rigorous evaluation free from data leakage, we re-evaluated all models on a strictly de-duplicated subset of the test set. We removed from DAS testing dataset any RNA targets that appeared in the training sets of baseline models and then evaluate those baseline models on the rest. The results are reported in Table 7.

Table 7: Performance comparison on the strict De-duplicated Test Set. RiboPO achieves the best balance of structural fidelity (RMSD) and thermodynamic stability (MFE) without relying on data leakage.

| Model | Seq Rec | scMCC (2D) (↑) | RMSD (Å) (↓) | %RMSD ≤ 8Å (↑) | TM-score (↑) | MFE (kcal/mol) (↓) |
|---|---|---|---|---|---|---|
| RDesign | 0.42 | 0.21 | 16.48 | 0.23 | 0.13 | -22.06 |
| R3Design | 0.51 | 0.59 | 14.52 | 0.47 | 0.27 | -26.57 |
| RhoDesign | 0.47 | 0.21 | 14.82 | 0.18 | 0.14 | -13.53 |
| RiboDiffusion | 0.52 | 0.49 | 13.52 | 0.25 | 0.22 | -20.28 |
| gRNAde (Base) | **0.53** | 0.61 | 11.43 | 0.29 | 0.29 | -32.84 |
| **RiboPO (Round 2)** | 0.50 | **0.70** | **10.23** | **0.51** | **0.31** | **-36.86** |

## A.5 Pareto Frontier Analysis of Multi-Objective Trade-offs

To analyze the trade-off between structural fidelity, thermodynamic stability, and sequence recovery, we performed a Pareto analysis across training rounds and varying $\beta$ hyperparameters. Figure 3 illustrates the movement of the model in the multi-objective space.

**Analysis.**

- **Designability Frontier (Figure 3B):** The base model (gRNAde, square) sits in a sub-optimal region. RiboPO Round 2 (Star) pushes the frontier significantly towards the top-left (Higher scMCC, Lower RMSD), indicating an expanded "designable basin." Later rounds (R3-R5) continue to improve MFE but show diminishing returns on structural consistency.

- **Identity vs. Stability (Figure 3A):** We observe a slight drop in sequence recovery ($\sim 3\%$) correlated with a massive improvement in MFE ($12.3\%$). This confirms that the model is exploring non-native, thermodynamically favorable regions of the sequence space rather than failing to learn.

## A.6 Cross-Tool Robustness Evaluation

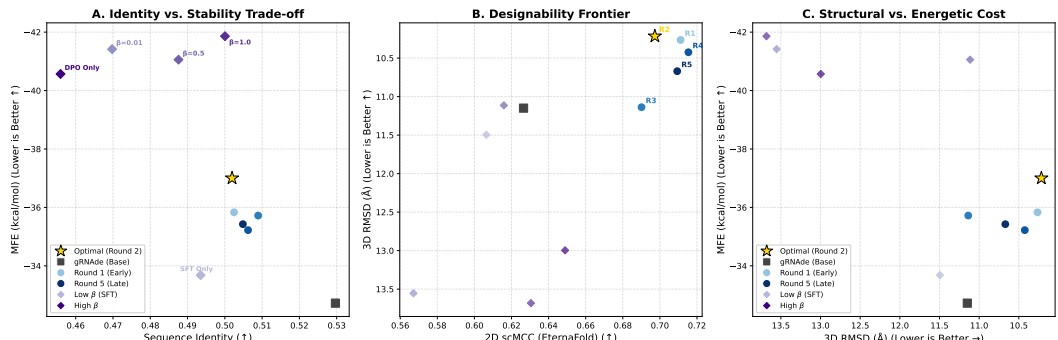

Figure 3: **Pareto Frontier Analysis.** (A) Sequence Identity vs. MFE trade-off. (B) Designability Frontier (scMCC vs. RMSD). (C) Structural vs. Energetic Cost. RiboPO (Round 2, Star) represents the optimal operating point.

To address concerns that RiboPO might be overfitting to the specific energy function used during training (ViennaRNA), we evaluated the generated sequences using two external thermodynamic engines: NUPACK (Fornace et al., 2022) and RNAstructure (Bellaousov et al., 2013).

As shown in Table 8, the stability improvements generalize well. RiboPO sequences show a 10.15% improvement in MFE under NUPACK and a 12.32% improvement under RNAstructure compared to the baseline. This confirms that the method learns generalized thermodynamic stability principles rather than tool-specific artifacts.

Table 8: Cross-Tool Robustness. MFE improvements generalize to NUPACK and RNAstructure engines, validating biophysical robustness.

| Method | *NUPACK* MFE | *RNAstructure* MFE | Improv. (*NUPACK / RNAstructure %*) |
|---|---|---|---|
| gRNAde (Base) | -28.90 | -30.53 | - |
| RiboPO (Round 2) | -31.84 | -34.29 | **+10.2% / +12.3%** |

**3D Structural Generalization.** We assessed the structural fidelity of sequences generated by the gRNAde base model and RiboPO (Round 2) using AlphaFold3 (Abramson et al., 2024) and DR-Fold2 (Li et al., 2025). These tools employ folding algorithms distinct from the RhoFold+ engine used in our reward loop, serving as different proxies.

As shown in Table 9, RiboPO consistently outperforms the baseline across all independent evaluators. Notably, evaluation with AlphaFold3—currently considered the gold standard in biomolecular structure prediction—reveals that RiboPO designs achieve lower RMSD (2.81 Å vs. 3.00 Å) and higher TM-scores (0.43 vs. 0.41) compared to the baseline. This improvement on unseen predictors indicates that the RLPF framework steers the policy toward universally valid structural basins rather than merely exploiting artifacts of the training surrogate.

It's also worth-noting that AlphaFold3 and DRFold2 output better self-consistency scores on the de-duplicated dataset, but performed complementary results on individual samples (Martinović et al., 2024), indicating that the RNA 3D structure prediction hasn't arrived the "AlphaFold Moment" (Li et al., 2025). We'd like to call for the development of better RNA structure prediction models by the structure prediction community, and to claim that with better 3D structure prediction tools, our RiboPO/RLPF can have less biased feedback and output more meaningful RNA designs.

### A.7 NATIVE STRUCTURE EVALUATION BENCHMARK

To contextualize the performance of all models, we report the evaluation metrics on the native ground-truth sequences in Table 10. These values represent the "upper bound" (or target) for current predictors (RhoFold/EternaFold).

Table 9: **3D Cross-Tool Robustness.** Evaluation of generated sequences using independent structure predictors (AlphaFold3 and DRFold2) not seen during training. Across all the mentioned tools, RiboPO (Multi-Round 2) demonstrates consistent structural improvements over the baseline, validating that the geometric gains are model-agnostic.

| Model | AlphaFold 3 | | | DRFold 2 | |
|---|---|---|---|---|---|
| | RMSD ($\downarrow$) | TM-Score ($\uparrow$) | pLDDT ($\uparrow$) | RMSD ($\downarrow$) | TM-Score ($\uparrow$) |
| gRNAde (Base) | 3.00 | 0.41 | 61.02 | 2.96 | 0.37 |
| **RiboPO (Round 2)** | **2.81** | **0.43** | **62.05** | **2.79** | **0.39** |

Table 10: Evaluation of Native Sequences (Ground Truth). Note that non-zero RMSD arises from the use of structure predictors (RhoFold) on native sequences. These values serve as the upper bound for current prediction tools.

| Sequence | Seq Rec | scMCC | RMSD (Å) | %RMSD $\leq 8$Å | TM-score | %TM $\geq 0.45$ | pLDDT | %pLDDT $\geq 0.70$ | MFE | ED/nt |
|---|---|---|---|---|---|---|---|---|---|---|
| Native | 1.00 | 0.86 | 5.78 | 0.82 | 0.56 | 0.65 | 0.74 | 0.72 | -42.83 | 0.002 |

## A.8    STATISTICAL SIGNIFICANCE AND VARIANCE ANALYSIS

To ensure that the reported improvements are robust and not artifacts of random initialization, we evaluated the optimal RiboPO model (Round 2) and the baseline gRNAde across 4 independent random seeds $(0, 6, 40, 42)$. Table 11 reports the mean and standard deviation ($\mu \pm \sigma$) for key performance metrics.

RiboPO demonstrates consistent superiority in thermodynamic stability (MFE: $-36.98$ vs. $-32.71$) and structural consistency (scMCC: $0.69$ vs. $0.62$) with low variance. Crucially, we observe statistically significant gains in tertiary structure confidence (pLDDT: $0.65$ vs. $0.63$), validating that the preference optimization improves the density of high-quality designs.

Table 11: Variance Analysis across 4 random seeds ($N = 4$). Values represent Mean $\pm$ Standard Deviation. RiboPO achieves statistically significant improvements across both structural and thermodynamic axes.

| Model | Seq Rec | scMCC | RMSD (Å) $\downarrow$ | TM-score $\uparrow$ | pLDDT $\uparrow$ | MFE (kcal/mol) $\downarrow$ |
|---|---|---|---|---|---|---|
| gRNAde (Base) | $0.531 \pm 0.004$ | $0.625 \pm 0.015$ | $11.17 \pm 0.22$ | $0.297 \pm 0.003$ | $0.626 \pm 0.004$ | $-32.71 \pm 0.12$ |
| **RiboPO (R2)** | $0.502 \pm 0.001$ | $\mathbf{0.694 \pm 0.007}$ | $\mathbf{10.31 \pm 0.25}$ | $\mathbf{0.307 \pm 0.009}$ | $\mathbf{0.649 \pm 0.004}$ | $\mathbf{-36.98 \pm 0.13}$ |

## A.9    THEORETICAL ANALYSIS

In this subsection we provide a theoretical perspective on RiboPO/RLPF. We (i) formalize our preference construction in terms of an underlying multi-objective physical reward and an "ideal" Boltzmann distribution over stable sequences, (ii) connect the DPO training objective with a *fixed reference policy* to a KL-regularized variational problem in which the base model acts as a prior, (iii) relate the multi-round training scheme to iterative refinement of this objective and movement into designable basins, and (iv) discuss multi-objective trade-offs, Goodhart-type effects, and the role of biased surrogate predictors and the SFT anchor.

### A.9.1    LATENT MULTI-OBJECTIVE REWARD AND BOLTZMANN VIEW

For each backbone $G$, we associate a sequence $s$ with a vector of normalized physical features

$$\phi(s, G) \in \mathbb{R}^d, \tag{8}$$

including tertiary-structure quality (e.g. RMSD, pLDDT), secondary-structure self-consistency (e.g. scMCC), and thermodynamic proxies (e.g. MFE, ensemble defect, target-structure probability, entropy, melting temperature), as defined in the main text. We normalize and, when necessary, sign-flip individual metrics so that larger values of each component of $\phi(s, G)$ correspond to better structural fidelity or stability. We then assume there exists a latent *multi-objective* reward function

$$r(s, G) = w^{\top} \phi(s, G), \qquad w \succeq 0, \tag{9}$$

that encodes the relative importance of structural fidelity and thermodynamic stability in RNA inverse folding.

A particularly interpretable special case is obtained by defining a composite physical energy function

$$E(s, G) = \lambda_{\text{thermo}} \, E_{\text{thermo}}(s) + \lambda_{\text{geom}} \, E_{\text{geom}}(s, G), \tag{10}$$

where $E_{\text{thermo}}$ aggregates thermodynamic terms (e.g. MFE and ensemble measures) and $E_{\text{geom}}$ penalizes structural deviations (e.g. RMSD). Setting $r(s, G) = -E(s, G)$ recovers a standard Boltzmann form.

Let $p^*(s \mid G)$ denote an "ideal" physical distribution over sequences for backbone $G$, defined by a Boltzmann distribution

$$p^*(s \mid G) \; \propto \; \exp\{-\beta E(s, G)\} \; = \; \exp\{\beta r(s, G)\}, \tag{11}$$

for some inverse temperature $\beta > 0$. In our setting, we also have a strong backbone-conditioned autoregressive base model $\pi_{\text{ref}}(s \mid G)$ (gRNAde), which we can view as a data-driven prior over plausible sequences for $G$. Combining the prior with the energy-based likelihood yields a target distribution

$$q^*(s \mid G) \; \propto \; \pi_{\text{ref}}(s \mid G) \, \exp\{\beta r(s, G)\}. \tag{12}$$

Thus, an ideal optimizer would transform the base model $\pi_{\text{ref}}$ into a posterior-like policy $q^*$ that favors sequences with high structural and thermodynamic reward while remaining close to the data manifold learned by $\pi_{\text{ref}}$.

Our preference construction is designed to be consistent with such a reward. We first apply a *feasibility gate* that enforces that a candidate winner $s_w$ lies in a physically plausible region:

$$\text{pLDDT}(g(s_w)) > \kappa, \qquad \text{RMSD}(G, g(s_w)) < \gamma, \tag{13}$$

where $g(\cdot)$ is the structure predictor and $\kappa, \gamma$ are thresholds. Second, for each feasible pair $(s_w, s_l)$ we enforce margin conditions on structural deltas:

$$\text{pLDDT}(s_w) - \text{pLDDT}(s_l) \geq \gamma_r \sigma_{\text{pLDDT}}, \tag{14}$$

$$\text{RMSD}(s_l) - \text{RMSD}(s_w) \geq \gamma_r \sigma_{\text{RMSD}}, \tag{15}$$

where $\sigma_m$ is the empirical scale of metric $m(\cdot)$ and $\gamma_r$ controls how strict the preference is in round $r$. In the implementation (Fig. 1), these filters and margins are applied only to the structural metrics pLDDT and RMSD; thermodynamic metrics such as MFE enter through the latent reward $r(s, G)$ and downstream evaluation but do not receive explicit margins. These conditions exclude candidates unlikely to fold correctly and filter out near-equivalent pairs where metric differences are on the same order as predictor noise, improving the signal-to-noise ratio of the implicit reward. Normalization by $\sigma_m$ renders metrics on different scales commensurate and stabilizes the DPO likelihood-ratio updates.

To connect this construction with the latent reward in equation 9, we adopt the following idealized assumption: there exists a non-negative weight vector $w \succeq 0$ such that

$$r(s_w, G) - r(s_l, G) \; \geq \; \Delta_r \; > \; 0 \tag{16}$$

for all retained pairs $(G, s_w, s_l)$. In the special case $r = -E$, this assumption is equivalent to requiring $E(s_w, G) < E(s_l, G)$ by a margin, i.e. that the winner is stochastically more stable.

We can therefore model our preferences using a standard Bradley–Terry–Luce (BTL) logistic model. Let $s_w \succ s_l$ denote the event that $s_w$ is preferred over $s_l$. We posit that, conditional on $G$,

$$\mathbb{P}(s_w \succ s_l \mid G) \; = \; \sigma\big(\beta\big(r(s_w, G) - r(s_l, G)\big)\big), \tag{17}$$

where $\sigma$ denotes the logistic sigmoid and $\beta > 0$ is an inverse-temperature parameter. In the idealized noiseless case where equation 16 holds, the empirical preferences are consistent with a single latent reward $r$ (or equivalently a single energy function $E$ via $r = -E$).

### A.9.2 DPO with a fixed reference as KL-regularized variational inference

Let $\pi_\theta(s \mid G)$ denote the policy parameterized by RiboPO, and $\pi_{\text{ref}}(s \mid G)$ the fixed reference policy (the base gRNAde model). For each preference triple $(G, s_w, s_l)$, the DPO loss used in RiboPO is

$$L_{\text{DPO}}(\theta) = -\mathbb{E}_{(G, s_w, s_l)} \left[ \log \sigma \Big( \beta \Big( \log \pi_\theta(s_w \mid G) - \log \pi_\theta(s_l \mid G) - \log \pi_{\text{ref}}(s_w \mid G) + \log \pi_{\text{ref}}(s_l \mid G) \Big) \Big) \right]. \tag{18}$$

We additionally apply a supervised fine-tuning (SFT) anchor on the winners:

$$L_{\text{SFT}}(\theta) = -\mathbb{E}_{(G, s_w)}\big[\log \pi_\theta(s_w \mid G)\big], \tag{19}$$

and train RiboPO with the combined objective

$$L(\theta) = L_{\text{DPO}}(\theta) + \lambda_{\text{SFT}} L_{\text{SFT}}(\theta), \tag{20}$$

where $\lambda_{\text{SFT}} \geq 0$ controls the strength of the anchor.

Although the optimization problem we implement is written entirely in terms of the preference cross-entropy equation 18 and the SFT term equation 19, the *population optimum* of the DPO part admits a useful variational characterization. Under the BTL model equation 17 with reward $r(s, G)$ and in the infinite-data, realizable limit, the unique policy $\pi^*$ that exactly matches the induced preferences can be written in closed form as

$$\pi^*(s \mid G) \; \propto \; \pi_{\text{ref}}(s \mid G) \exp\big\{\beta r(s, G)\big\}. \tag{21}$$

Comparing equation 21 with equation 12 shows that $\pi^*$ coincides with the target posterior-like distribution $q^*$. Equivalently, $\pi^*$ is the unique minimizer of the KL divergence

$$\pi^* \; = \; \arg\min_\pi \; \mathbb{E}_G\Big[\text{KL}\big(\pi(\cdot \mid G) \,\|\, q^*(\cdot \mid G)\big)\Big], \tag{22}$$

and the unique maximizer of the KL-regularized reward objective

$$J(\pi) = \mathbb{E}_G\left[\mathbb{E}_{s \sim \pi(\cdot \mid G)}\big[r(s, G)\big] - \frac{1}{\beta}\text{KL}\big(\pi(\cdot \mid G) \,\|\, \pi_{\text{ref}}(\cdot \mid G)\big)\right]. \tag{23}$$

Thus, while our training loss is expressed in terms of pairwise preference logistic losses, the corresponding population optimum can be interpreted as the solution to a KL-regularized physical reward maximization problem, or equivalently as variational inference that projects the base model $\pi_{\text{ref}}$ onto the physically informed target $q^*$. These characterizations hold at the population level under the modeling assumptions above; in practice we optimize a parametric policy $\pi_\theta$ using finitely many preference pairs, which yields an approximation to this ideal solution.

The fact that $\pi_{\text{ref}}$ is kept *fixed* is crucial for this interpretation: the KL term in equation 23 is always taken with respect to the same prior $\pi_{\text{ref}}$, so the objective $J(\pi)$ does not change over rounds. If the reference policy were instead updated to the current $\pi_\theta$ at each step, the center of the KL penalty would move, effectively weakening the prior over time and making it easier for the policy to drift toward reward-hacking regimes that exploit imperfections in the surrogate metrics. In contrast, a fixed $\pi_{\text{ref}}$ maintains a consistent trade-off between improving the physical reward and staying close to the data-driven prior over biologically plausible sequences.

For intuition, consider the derivative of $L_{\text{DPO}}$ with respect to the log-probabilities of the winner and loser. Let

$$z = \beta \left(\log \frac{\pi_\theta(s_w \mid G)}{\pi_\theta(s_l \mid G)} - \log \frac{\pi_{\text{ref}}(s_w \mid G)}{\pi_{\text{ref}}(s_l \mid G)}\right). \tag{24}$$

Then

$$\frac{\partial L_{\text{DPO}}}{\partial \log \pi_\theta(s_w \mid G)} = \beta\big(\sigma(z) - 1\big), \qquad \frac{\partial L_{\text{DPO}}}{\partial \log \pi_\theta(s_l \mid G)} = \beta\big(1 - \sigma(z)\big). \tag{25}$$

When $\pi_\theta$ under-prefers the winner relative to the reference ($z$ small), $\sigma(z) \ll 1$ and the gradient increases $\log \pi_\theta(s_w \mid G)$ while decreasing $\log \pi_\theta(s_l \mid G)$. When $\pi_\theta$ already strongly prefers $s_w$ over $s_l$, $\sigma(z) \approx 1$ and the gradient vanishes. This yields a well-behaved optimization landscape in which only "hard" or ambiguous preferences drive learning. The SFT term equation 19 then sharpens the mode of the learned policy around high-reward winners, mitigating reward hacking and mode collapse.

### A.9.3 MULTI-ROUND TRAINING AND DESIGNABLE BASINS

We now link this perspective to multi-round RLPF and the notion of designable basins. For a given backbone $G$, define the designable set of sequences

$$\mathcal{S}_{\text{des}}(G) = \{s : \text{RMSD}(G, g(s)) \leq \tau_{\text{struct}}, \; \text{scMCC}(s, G) \geq \tau_{\text{struct}}, \; \text{MFE}(s) \leq \tau_{\text{thermo}}\}, \tag{26}$$

for suitable structural and thermodynamic thresholds $\tau_{\text{struct}}, \tau_{\text{thermo}}$. The *designability* of $G$ under a policy $\pi$ is then

$$D_\pi(G) = \sum_{s \in \mathcal{S}_{\text{des}}(G)} \pi(s \mid G), \tag{27}$$

i.e., the total probability mass that $\pi$ assigns to sequences that fold accurately and occupy deep thermodynamic basins.

By construction, the feasibility gate equation 13 ensures that winners $s_w$ lie in a physically plausible region consistent with the structural constraints used in equation 26 (in particular, small RMSD); losers may lie inside or outside this broader set. Together with the additional structural requirement on scMCC and the thermodynamic requirement on MFE in equation 26, this defines a subset of sequences that we regard as designable for $G$. Under the modeling assumptions of the previous subsection and the gradient expressions in equation 25, updates therefore tend to *redistribute* probability mass from low-reward or infeasible sequences into $\mathcal{S}_{\text{des}}(G)$, so that $D_\pi(G)$ is expected to increase over training, at least on average. This provides a conceptual explanation for why RiboPO moves the policy toward more designable basins in sequence space.

The multi-round training scheme further refines this process. Each round $r$ proceeds in two steps:

1. Use the current policy $\pi_{\theta_{r-1}}$ to generate a pool of candidate sequences for each backbone $G$.

2. Construct preference pairs from this pool using the same gating and margin criteria, and update $\pi_{\theta_r}$ by minimizing equation 20 with *the same* reference policy $\pi_{\text{ref}}$.

From the KL-regularized perspective equation 23, each round can be viewed as performing an approximate mirror-descent step on $J(\pi)$ from the previous policy, but always with the KL term centered at the fixed prior $\pi_{\text{ref}}$. The candidate-generation step enriches the support over which preferences are constructed: early rounds primarily explore regions favored by $\pi_{\text{ref}}$, whereas later rounds can propose sequences in newly discovered high-reward basins. The fixed reference ensures that, despite this exploration, all rounds optimize the *same* trade-off between physical reward and fidelity to the base model.

Empirically, this picture is consistent with the observation that intermediate rounds often lie near the Pareto front between recovery, structural, and thermodynamic metrics: early rounds move the policy from a purely data-driven prior toward physically favorable basins; later rounds refine within those basins but may over-specialize to small and noisy differences in surrogate metrics, leading to diminishing returns or regressions on some objectives.

### A.9.4 MULTI-OBJECTIVE TRADE-OFFS, GOODHART EFFECTS, AND THE SFT ANCHOR

The latent reward $r(s, G)$ in equation 9 is explicitly multi-objective: it combines structural fidelity (RMSD, scMCC, pLDDT) with thermodynamic terms (MFE and ensemble metrics). Optimizing such a scalarized reward inevitably induces trade-offs: improving one component (e.g. MFE) may worsen others (e.g. exact sequence recovery). This is reflected in the empirical Pareto-front analyses, where different choices of $\beta$ and round index yield different operating points.

From a theoretical standpoint, over-optimizing a single proxy (e.g. MFE as a surrogate for true stability and biological plausibility) can lead to Goodhart-type failure modes: the optimizer exploits idiosyncrasies of the proxy and drifts toward unphysical, low-complexity structures that score well under the surrogate but are biologically implausible. Our framework mitigates this in three ways:

**Multi-objective reward.** First, preferences are only retained when the winner is at least as good on the structural metrics. This reduces the risk of collapsing onto thermodynamically trivial but structurally incorrect solutions, since the latent reward $r(s, G)$ upweights sequences that jointly exhibit strong structural fidelity and favorable thermodynamics.

**KL regularization against the base model.** Second, the KL-regularized objective equation 23 shows that, in the idealized limit, DPO with a fixed reference seeks policies that improve $r(s, G)$ while remaining close to the base model $\pi_{\text{ref}}$. This discourages pathological modes that deviate

drastically from the distribution of sequences seen in training, even if they exploit idiosyncrasies of the surrogate metrics.

**SFT as a distributional anchor.** Third, the SFT term equation 19 acts as an explicit *distributional anchor*. While the DPO component pulls the policy toward low-energy, structurally consistent sequences, the SFT component biases $\pi_\theta$ toward sequences similar to the winners produced by the base model. At a high level, the combined objective seeks a policy that (i) increases expected physical reward relative to $\pi_{\text{ref}}$ and (ii) stays on or near the manifold of sequences that are likely under a high-quality inverse folder. This theoretical balance explains the empirical trade-offs observed in our experiments: RiboPO is allowed to sacrifice a modest amount of exact sequence recovery to access regions of sequence space with improved structure and stability, but the KL and SFT anchors limit the extent to which the policy can exploit surrogate-specific artifacts.

### A.9.5 SURROGATE PREDICTORS, BIAS, AND EXTENSIBILITY

Finally, we address the fact that RiboPO relies on imperfect and potentially biased surrogate predictors of RNA structure and energetics, and argue that the framework remains principled and extensible as stronger models become available.

Let $r^*(s, G)$ denote an unknown "true" physical reward, e.g. derived from exact folding and thermodynamics or experimental measurements. We do not observe $r^*$ directly. Instead, we construct a surrogate reward $r_\Phi(s, G)$ based on a feature vector

$$\Phi(s, G) = \big(\phi^{\text{3D}}(s, G), \phi^{\text{2D}}(s, G), \phi^{\text{thermo}}(s, G), \dots \big), \tag{28}$$

which aggregates outputs from multiple biophysical tools (e.g. RhoFold+, EternaFold, ViennaRNA). Our working reward can be written as

$$r_\Phi(s, G) = w^\top \Phi(s, G), \tag{29}$$

for some non-negative weight vector $w$. Each component of $\Phi$ is a biased proxy for an underlying physical quantity, so we can decompose

$$r_\Phi(s, G) = r^*(s, G) + \eta(s, G), \tag{30}$$

where $\eta$ captures the combined bias and noise of the surrogate predictors and the chosen weights.

Our preference labels are generated by comparing $r_\Phi$: we declare $s_w$ preferred to $s_l$ if $r_\Phi(s_w, G) > r_\Phi(s_l, G)$ and the gating conditions hold. Let $s_w \succ^* s_l$ denote the "true" preference according to $r^*$. We assume that, for pairs surviving the gate, the surrogate and true preferences are aligned with probability at least $1 - \rho$:

$$\mathbb{P}\Big(\text{sign}\big(r_\Phi(s_w, G) - r_\Phi(s_l, G)\big) = \text{sign}\big(r^*(s_w, G) - r^*(s_l, G)\big)\Big) \geq 1 - \rho, \qquad \rho < \tfrac{1}{2}. \tag{31}$$

Intuitively, even though individual predictors are imperfect, they rank higher-quality sequences above clearly worse ones more often than chance. Under this mild assumption, the expected gradient of the DPO objective is still correlated with the gradient of the true reward $r^*$: in aggregate, preference updates move the policy toward better $r^*$, albeit with some variance due to mis-specified or noisy labels.

Crucially, the use of a *vector* of surrogate features $\Phi(s, G)$ rather than a single tool reduces the risk of overfitting to any one predictor's bias. Only sequences that look consistently better across multiple, differently biased views (3D, 2D, thermodynamics) tend to survive the gating and margin filters and be chosen as winners. This multi-tool formulation is modular: as stronger predictors of RNA structure and energetics become available, they can be incorporated into $\Phi$ (or used to replace existing components) without changing the RLPF/RiboPO learning algorithm.

In the idealized limit where $\sup_{s,G}|r_\Phi(s, G) - r^*(s, G)| \to 0$ as tools improve, the mis-ranking probability $\rho$ in equation 31 tends to zero and the induced preferences converge to the true physical preferences. In this limit, DPO trained on preferences from $r_\Phi$ converges (in function space) to the same KL-regularized optimal policy one would obtain by training directly on $r^*$. Thus, the proposed RLPF/RiboPO framework can be viewed as a modular pipeline whose performance is guaranteed to systematically improve as biophysical surrogates become more accurate, while already yielding meaningful improvements under today's imperfect RNA structure and thermodynamics models.

## A.10 VISUAL CASE STUDIES

To qualitatively demonstrate the impact of thermodynamic preference optimization, we present a representative case study from the test set comparing the native structure against designs from the baseline (gRNAde) and our method (RiboPO).

**Secondary Structure Topology.** Figure 4 illustrates the secondary structure landscape of the sample $4MEG\_1\_B$ by forna (Kerpedjiev et al., 2015). The baseline model, gRNAde, suffers from a severe topological hallucination, predicting a complex four-way junction (cloverleaf-like architecture) instead of the native extended linear conformation. This error likely arises because the baseline optimizes geometric proxies without sufficient penalties for ensemble instability, causing the sequence to settle into a false energetic minimum. In contrast, RiboPO successfully recovers the correct global topology, accurately separating the 5' and 3' domains with the appropriate central linker region. This validates that the inclusion of thermodynamic feedback ($f_{thermo}$) effectively constrains the search space to biologically viable secondary structures.

**Tertiary Structure Fidelity.** Figure 5 extends this analysis to the tertiary level. We used PyMol (Schrödinger & DeLano) to visualized the structure (predicted by AlphaFold3 (Abramson et al., 2024)) superposition of the sample $5C7W\_1\_C$. The gRNAde prediction exhibits significant deviations from the native backbone, particularly in the orientation of the helical domains, resulting in a high RMSD and potential steric clashes. The RiboPO design maintains a tighter global superposition with the native structure. By penalizing physical inconsistencies during the reinforcement learning phase, RiboPO corrects these domain orientation errors, yielding a structure that is both geometrically accurate and thermodynamically stable.

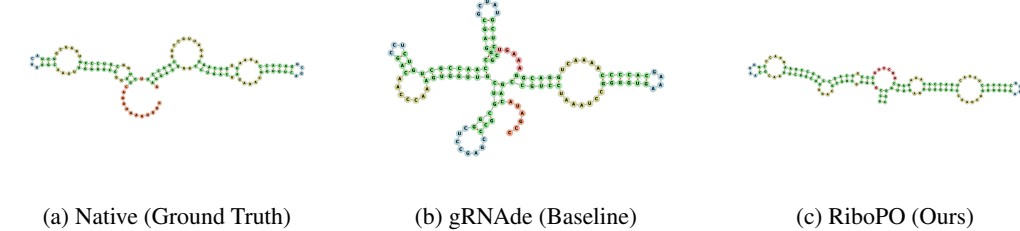

(a) Native (Ground Truth)          (b) gRNAde (Baseline)          (c) RiboPO (Ours)

Figure 4: Comparison of 2D Secondary Structures. While gRNAde hallucinates a complex multiway junction, RiboPO correctly identifies the native linear topology, demonstrating superior designability.

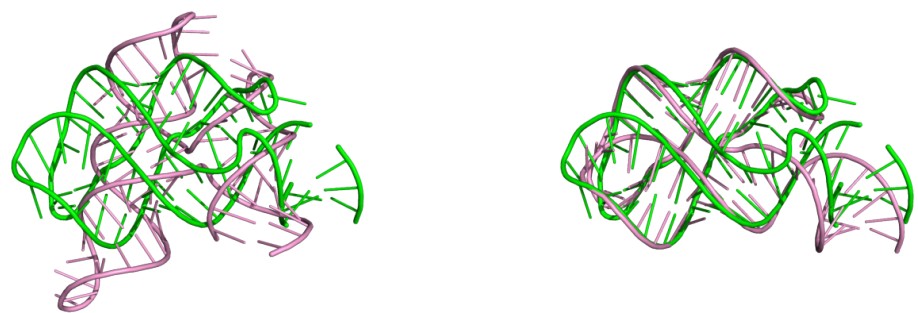

(a) gRNAde vs. Native (Seq Rec. = 0.446, TMscore = 0.060, RMSD = 18.455)          (b) RiboPO vs. Native (Seq Rec. = 0.446, TMscore = 0.462, RMSD = 3.684)

Figure 5: 3D Structure Superposition (Green: Native, Pink: Designed). RiboPO achieves significantly better alignment and domain orientation compared to the baseline, correcting the global fold errors.

### A.11 LLM USAGE

We utilized a large language model to assist with improving the grammar, spelling, and overall clarity of the manuscript. The scientific contributions, including all ideas, methods, and results, are our own. We take full responsibility for the content of this paper.

