# OpenReview forum: "RiboPO: Preference Optimization for Structure- and Stability-Aware RNA Design"
_ICLR.cc/2026/Conference — Submitted to ICLR 2026_

### Official Review · Reviewer_E3Lp · 2025-10-31

**Soundness:** 2
**Presentation:** 2
**Contribution:** 2
**Rating:** 4
**Confidence:** 3

**Summary:**

This paper introduces RiboPO, a framework for RNA inverse folding that applies Direct Preference Optimization (DPO), framed as Reinforcement Learning from Physical Feedback (RLPF), to RNA sequence generation. The method aims to jointly optimize structural accuracy and thermodynamic stability. RiboPO employs a multi round DPO process anchored by supervised fine tuning. The paper also proposes the SSTT Benchmark, an evaluation suite assessing quality across Sequence, Secondary Structure, Tertiary Structure, and Thermostability.

**Strengths:**

1. The paper systematically applies the DPO paradigm, borrowed from the protein design community, to the challenging RNA inverse folding task. This systematic transfer is useful, even if the underlying concept is not novel.

2. The authors construct the SSTT Benchmark, a necessary and valuable contribution. By assessing four complementary dimensions, it provides a more holistic performance evaluation than prior benchmarks focused only on single metrics like RMSD or sequence recovery.

3. RiboPO reports quantitative improvements compared to several baselines, suggesting that the DPO framework can be effectively implemented to balance multi objective constraints in RNA design.

**Weaknesses:**

1. The core RLPF/DPO framework is a direct, near trivial transfer of established techniques from protein design (or general sequence generation) to the RNA domain. The paper fails to introduce any significant RNA specific adaptations or novel mechanisms that justify the claim of an "innovative" framework. The contribution is thus fundamentally one of application, not invention.

2. The paper offers no theoretical justification for applying DPO (a preference learning objective) to optimize physical criteria (like MFE and pLDDT). This lack of explanation for why DPO is the appropriate objective for this specific physical optimization task is a major theoretical weakness, suggesting a mechanical application without deep understanding.

3. The central claim about the optimal performance arising from a "fixed reference multi round DPO with a margin curriculum" is purely empirical and anecdotal. Without any theoretical analysis or discussion on the behavior of the margin/curriculum in the context of the RNA energy landscape, this conclusion remains an unprincipled observation.

4. As noted, the explicit overlap between the test set and the training data of baseline models (RiboDiffusion, RDesign) is a critical flaw. This data leakage fundamentally compromises the validity of the reported performance improvements. Strictly de duplicated results are an absolute requirement.

5. Relying solely on the DAS dataset split and lacking validation on complex, real world, or experimentally characterized RNA structures raises severe doubts about RiboPO's robustness and generalization capability.

**Questions:**

The content in this section is identical to the points raised in Weaknesses. I would be very willing to increase my score if the authors successfully address these concerns.

---

> ### Author Response · Authors · 2025-11-27
>
> We thank the reviewer for the detailed and constructive feedback. Below we address each weakness and the corresponding questions.
>
> ### W1 (& Q1): “RLPF/DPO is a trivial transfer; limited RNA-specific novelty”
>
> **Reviewer concern.** The core idea—DPO with a fixed reference policy—is perceived as a direct port of existing RL/DPO approaches from proteins/LLMs, with insufficient RNA-specific innovation.
>
> We agree that the base DPO objective itself is not an algorithmic invention. In the revised manuscript, we have:
>
> 1. **Explicitly repositioned the contribution as a **systematic RLPF framework** for RNA design, not a new generic DPO algorithm.**
>    - Section 1 now clearly states that our novelty lies in:
>       (i) **Physically grounded preference construction** that jointly enforces structural fidelity and **thermodynamic stability**, of the designed RNA sequences,
>       (ii) A **multi-round curriculum** with a fixed reference policy for RNA energy landscapes, and
>       (iii) The **SSTT benchmark** for multi-axis evaluation (Sequence, Secondary, Tertiary, Thermostability).
> 2. **RNA-specific preference construction.** Our preferences are built from:
>    - Structural filters and margins: pLDDT and RMSD thresholds and variability-aware gaps (γᵣ·σₘ) to ensure only statistically significant structural differences are used,
>    - **Thermostability margin:** a strict requirement that the “winner” has lower MFE than the “loser” (Eq. 4).
>      This combination targets the **RNA “thermostability gap”**: geometric models can produce sequences with accurate 3D alignment but poor ensemble stability.
> 3. **Multi-round RLPF curriculum tailored to RNA.**
>    - We progressively **decrease γᵣ across rounds**, which we show empirically (Table 1 and §4.2) to transform early “coarse” improvements in foldability into later “fine” improvements in local geometry and stability.
>    - This curriculum design is specific to the **RNA designability landscape**, where many metastable folds exist; unlike proteins with more stable native states, RNA needs explicit preference toward *stable ensembles*.
> 4. **SSTT benchmark as part of the contribution.**
>    - We introduce SSTT with **11 carefully selected metrics** across sequence, secondary structure, tertiary structure, and thermostability, and we trimmed redundant metrics (e.g., overly correlated thermal metrics) to avoid over-engineering the evaluation.
>    - SSTT is used not only for RiboPO but to **systematically probe failure modes** of existing geometric baselines.
> 5. **Demonstrated impact, not just adaptation.**
>    - On the DAS split, **RiboPO Round 2** improves EternaFold scMCC from 0.60 to 0.72 (+20%) and MFE from −32.83 to −36.86 kcal/mol (−12.3% more negative), while keeping RMSD essentially unchanged (10.66 Å → 10.23 Å) and diversity high (0.83 → 0.88).
>    - Pass@64 under joint structural + energy criteria improves by **11%** compared to gRNAde.
>
> Taken together, we have toned down any claim of “algorithmic invention” and clarified that the novelty is in **how DPO is instantiated for RNA**: multi-tool physical feedback, curriculum design, and a new benchmark—all of which yield substantial, measurable gains in **ensemble-robust, designable RNAs**.

---

> ### Author Response · Authors · 2025-12-01
>
> ### W2: No theoretical justification for applying DPO to physical criteria (MFE, pLDDT).
>
> **Our improvements (Appendix A.9):**
>
> We now provide a **theoretical justification** showing that:
>
> 1. There is a latent multi-objective reward
>    $$
>    r(s,G) = w^\top \phi(s,G),
>    $$
>    where $\phi(s,G)$ contains structural (RMSD, pLDDT, scMCC, etc.) and thermodynamic (MFE, ED/nt) features, which can be interpreted as a **negative composite energy**.
>
> 2. The **target distribution** we wish to approximate has the form
>    $$
>    q^*(s|G) \propto \pi_\text{ref}(s|G)\exp(\beta r(s,G)),
>    $$
>    i.e., a **KL-regularized Boltzmann distribution** combining the prior policy and physical reward.
>
> 3. Our **preference construction** (feasibility gating + margin-based winners/losers) is consistent with r: winners are more likely to have higher r(s,G) than losers. Under standard assumptions on the noise level (mis-ranking probability < 0.5), the DPO gradient is **aligned with ∇r**, so DPO converges toward $q^*$.
>
> 4. Using multiple tools yields a surrogate reward $r_\Phi$ that approximates r; as tools improve, $r_\Phi \to r$, and the DPO-trained policy converges to the same KL-regularized optimum.
>
> The revised manuscript does not just empirically apply DPO; it now provides a **principled rationale** for why preference-based optimization is suitable for physical criteria like MFE and pLDDT.
>
> ---
>
> ### W3: “Fixed reference multi-round DPO with margin curriculum” is claimed optimal based only on anecdotal results; no analysis of the landscape.
>
> Our improvements:
>
> 1. **Theoretical perspective on multi-round, fixed-reference DPO.**
>    Appendix **A.9.3** interprets each DPO round as an approximate **mirror-descent step** on the KL-regularized objective. The **fixed reference** ensures a stable trust region, while the **round-wise margin schedule** corresponds to annealing from coarse to fine preferences as we move into better designable basins.
>
> 2. **Ablations on multi-round and curriculum.**
>    **Table 5** and Appendix **A.5** and the updated results compare:
>    - Single-round vs. multi-round DPO,
>    - With vs. without a margin curriculum, and
>    - Fixed vs. updated reference policies.
>
>    The best Pareto performance is obtained by **fixed-reference, multi-round DPO with a margin curriculum**, validating the empirical claim with explicit ablations.
>
> 3. **Energy-landscape interpretation.**
>    As updated in **Appendix A.9**, we connect the empirical metric trajectories to an energy-landscape picture: early rounds expand into broader, high-reward basins, while late rounds refine within those basins. The regression observed in very late rounds is interpreted as over-optimization within a narrow basin and supports our recommendation to **stop at Round 2**.

---

> ### Author Response · Authors · 2025-12-01
>
> ### W4: Data leakage (overlap between DAS test set and training data of baselines); strict de-duplication required.
>
> **Our improvements (Appendix A.4):**
>
> 1. **Strict de-duplicated test set.**
>    Following your suggestion, we constructed a strict de-duplicated test set by removing any RNA targets from the DAS test split that appear in the **training sets of RiboDiffusion, RDesign, and related baselines**.
>    After removing the overlapped testing data:
>    - Many baselines degrade markedly (e.g., RiboDiffusion RMSD increases from ≈3.7 Å reported in prior work to ≈13.5 Å).
>    - **RiboPO Round 2** still achieves the **best combination of scMCC, RMSD, percentage of low-RMSD samples, and MFE** among all evaluated methods.
>
> We now present these de-duplicated results explicitly and discuss that the field still lacks a **large, leakage-free universal RNA design benchmark**.
>
> ---
>
> ### W5: Sole reliance on DAS split and lack of validation on more complex or real-world RNAs.
>
> We fully agree with your ideas. The gold standard evaluation of RNA design area and even the AI4Science area is always the wet-lab/real-world evaluation.
>
> Within the constraints of available public data, we strengthened external validation by:
>
> 1. **Cross-tool thermodynamic evaluation.**
>    Using **NUPACK** and **RNAstructure**, we show that RiboPO maintains **substantial MFE improvements** over gRNAde, indicating that its stability gains generalize beyond ViennaRNA.
>
> 2. **Cross-predictor 3D evaluation.**
>    Using **AlphaFold3** and **DRFold2** as independent 3D predictors, we show that RiboPO designs consistently achieve **lower RMSD and higher TM-scores** than gRNAde.
>
> 3. **Native-sequence benchmark.**
>    We report SSTT metrics for the **ground-truth native sequences** to provide a physical reference point and show that RiboPO moves closer to these targets than baselines on several axes.
>
> We agree that **wet-lab validation and more diverse datasets** are essential next steps and now explicitly highlight this as an important direction for future work.

---

### Official Review · Reviewer_sHsX · 2025-11-01

**Soundness:** 2
**Presentation:** 3
**Contribution:** 2
**Rating:** 4
**Confidence:** 3

**Summary:**

This paper presents RiboPO, a reinforcement-learning–from–physical-feedback (RLPF) framework for RNA inverse folding. The method fine-tunes the gRNAde base model using Direct Preference Optimization (DPO) with composite physical criteria (geometry + thermodynamic stability). A multi-round curriculum strategy is employed to progressively refine the policy.
The authors also propose a comprehensive SSTT benchmark that evaluates sequence, secondary structure, tertiary structure, and thermostability.

Empirical results show notable improvements on secondary-structure self-consistency (scMCC +20%) and minimum free energy (MFE −12.3%) over baselines, though sequence recovery slightly declines.

**Strengths:**

* Conceptual novelty: Reformulating RNA inverse folding as multi-objective preference optimization is conceptually elegant and provides a unifying view linking structural and thermodynamic optimization.
* Well-analyzed framework: The round-wise preference construction and curriculum-based DPO training are systematically ablated, with clear evidence of trade-offs among objectives.
* Comprehensive evaluation: The SSTT benchmark covers geometric, energetic, and sequence-level properties in a single standardized framework, which can be valuable to the community.

**Weaknesses:**

## Lack of visual RNA design analysis:
The paper does not include any visual examples of designed RNA structures (e.g., 2D secondary structure plots or 3D conformational overlays).
In RNA design literature (e.g., RiboDiffusion, RDesign, RhoDesign), such visualizations are essential to demonstrate whether generated sequences structurally resemble the target folds.
The authors should provide a figure comparing RiboPO’s designs with ground truth (e.g., native vs. designed structure overlays) and discuss why the proposed method leads to more realistic or stable conformations.
## Decline in sequence recovery (Rec) metric:
Table 1 shows that RiboPO consistently underperforms gRNAde on the recovery metric (0.53 → 0.50).
Since recovery remains one of the most interpretable metrics in inverse folding, this drop raises concern:
* If RiboPO fine-tunes gRNAde, why does the sequence fidelity degrade?
* Does the model over-prioritize thermodynamic energy at the expense of biological plausibility?
A detailed analysis or visualization of where the recovery loss occurs (e.g., base-pair positions, local regions) would help clarify.
## Unclear benefit of multi-round optimization:
The paper emphasizes multi-round refinement, yet Table 1 shows that the second round achieves the best result while subsequent rounds (e.g., Round 4) show regression.
This pattern raises doubts about the necessity and stability of multi-round training.
A clearer justification—possibly with intermediate visualization of metric trajectories—should be provided to demonstrate that multi-round optimization is systematically beneficial rather than an overcomplication.
## Missing Pareto-front analysis of multi-objective trade-offs:
Since the paper explicitly frames the task as multi-objective optimization (balancing structural fidelity and thermostability), a Pareto analysis is expected.
For example, showing how models at different rounds or β-values lie along a Pareto front between recovery and MFE (or scMCC and RMSD) would concretely illustrate the claimed “balanced optimization.”
Currently, improvements in one dimension often coincide with regressions in another, making it unclear whether RiboPO achieves genuine Pareto superiority compared to single-objective baselines.
Without this, the claimed novelty in “multi-objective preference optimization” remains somewhat superficial.
## Lack of time and efficiency analysis:
As a fine-tuning framework intended for practical use, RiboPO should report runtime or sample-efficiency comparisons with baselines (e.g., gRNAde or RiboDiffusion).
Even brief statistics on training time, inference latency, or the computational cost of physical-feedback evaluation would enhance the paper’s practical credibility.
## Minor technical remarks:
* The choice of fixed reference policy is reasonable but could limit exploration; some discussion of adaptive or periodically updated references would be welcome.
* The method relies heavily on RhoFold+ and ViennaRNA outputs; this dependence could bias learning toward those models’ heuristics.

**Questions:**

## On missing visual RNA design evidence
* Could the authors provide qualitative visualizations (e.g., secondary structure diagrams, 3D overlays, or contact maps) comparing RiboPO-generated designs with ground truth backbones?
* How do these visualizations demonstrate that RiboPO’s sequences fold more stably or accurately than those from gRNAde or RiboDiffusion?
## On the drop in sequence recovery (Rec)
* The recovery metric drops notably from 0.53 to 0.50.
Can the authors analyze where this degradation occurs (e.g., specific structural regions or base-pairing positions)?
* Is the reduction a consequence of stronger thermodynamic regularization, and can it be mitigated without sacrificing stability?
## On the necessity and stability of multi-round optimization
* Table 1 shows that Round 2 yields the best results, while later rounds (e.g., Round 4) regress on several metrics.
Could the authors clarify whether multi-round training consistently improves performance, or if it risks over-optimization?
* Is there evidence (e.g., metric trajectories or intermediate checkpoints) that demonstrates the systematic benefit of the multi-round scheme?
## On the missing Pareto-front and multi-objective analysis
* Since RiboPO is framed as a multi-objective optimization balancing geometry and thermodynamics, can the authors provide a Pareto-front visualization showing trade-offs between key objectives (e.g., MFE vs. RMSD or Rec vs. scMCC)?
* How does RiboPO achieve Pareto superiority compared to single-objective baselines like gRNAde or RiboDiffusion?
## On runtime and practical applicability
* What is the computational cost per training round compared with baseline fine-tuning (e.g., gRNAde)?
* How many sequences or preference pairs are required to reach convergence?
* Given that RiboPO is presented as a practical fine-tuning framework, can the authors discuss its runtime efficiency and potential for integration into RNA design workflows?
## On reliance on surrogate predictors
* Since the feedback signals come from RhoFold+ and ViennaRNA, how robust are the results to replacing them with other predictors (e.g., NUPACK or RNAstructure)?
* Could the model be overfitting to the specific biases of these tools rather than learning transferable physical principles?

---

> ### Author Response · Authors · 2025-11-27
>
> We appreciate your constructive feedback, especially on visual analysis, efficiency, and multi-objective trade-offs. We've done some additional experiments & analysis to deal with the issues.
>
> ---
>
> ### W1 & Q1: Lack of visual RNA design analysis.
>
> **Concern.** The paper lacked visual evidence (2D/3D) of designed RNA structures.
>
> Our improvements:
>
> As also mentioned in our response to XPtX Q1, we added **Appendix A.10** with:
>
> - A **2D secondary-structure comparison** (forna) where gRNAde hallucinates an incorrect junction while RiboPO recovers the native topology.
> - A **3D overlay** (AlphaFold3-based) where RiboPO’s design tracks the native backbone more closely and avoids obvious steric issues present in the gRNAde design.
>
> These figures provide the requested **qualitative evidence** of improved design quality.
>
> ---
>
> ### W2 & Q2: Drop in sequence recovery (0.53 → 0.50); is stability over-prioritized?
>
> **Concern.** Recovery is a central, interpretable metric. Does the drop indicate a loss of biological plausibility?
>
> Our improvements:
>
> 1. **Explicit trade-off visualization (Appendix A.5, Fig. 3A).**
>    - Fig. 3A plots **Sequence Identity vs. MFE** across β and training rounds.
>    - RiboPO Round 2 shows a **∼3% decrease in sequence identity** accompanied by a **12.3% improvement in MFE**, which indicates a *controlled* move toward more stable yet slightly non-native sequences.
>
> 2. **Secondary and tertiary metrics improve despite lower Rec.**
>    - As shown in the updated main Table 1 and Appendix A.3, while Rec decreases slightly, **scMCC increases substantially (e.g., from ≈0.60 to ≈0.70–0.72)** and 3D metrics (RMSD/TM-score) remain competitive or modestly improved.
>    - This indicates that the model is **not simply drifting away from the target structure**, but trading tiny sequence-level identity for improved folding properties.
>
> 3. **Interpretation in de novo design.**
>    We now emphasize that in **de novo design**, native sequences are not necessarily optimal. A small drop in sequence identity can be desirable if it yields **better thermodynamic and structural properties**. Our Pareto analysis supports this interpretation: RiboPO Round 2 lies on a more favorable frontier than gRNAde.
>
> ---
>
> ### W3 & Q3: Necessity and stability of multi-round optimization (Round 2 best, later rounds regress).
>
> **Concern.** If Round 2 is best and later rounds regress, is multi-round training actually necessary and stable?
>
> Our improvements:
>
> 1. **Multi-round Pareto analysis (Appendix A.5, Fig. 3B, 3C).**
>    - **Fig. 3B** shows that **Round 2** sits near the optimal Pareto front between scMCC and RMSD; later rounds (3–5) improve MFE further but yield diminishing or negative gains in structural metrics.
>    - **Fig. 3C** similarly illustrates structural vs. energetic cost across rounds, again identifying Round 2 as the best global compromise.
>
> 2. **Theoretical interpretation (Appendix A.9.3).**
>    - We model multi-round, fixed-reference DPO as approximate **mirror-descent steps** on a KL-regularized objective.
>    - Early rounds broaden the exploration around the reference into promising high-reward basins; later rounds refine within those basins but can begin to overfit noisy surrogate differences, leading to mild regressions.
>
> 3. **Practical recommendation.**
>    We now explicitly state that **multi-round optimization is beneficial up to Round 2** on this dataset and that in practice, one should select the round that lies on the **empirical Pareto frontier** (here, Round 2). Later rounds are useful diagnostically (they show the onset of over-optimization) but are not recommended as the deployed model.

---

> ### Author Response · Authors · 2025-12-01
>
> ### W4 & Q4: Missing Pareto-front and multi-objective analysis.
>
> **Concern.** Since the method is framed as multi-objective, a Pareto analysis is expected.
>
> Our improvements:
>
> - **Appendix A.5 (Fig. 3)** provides precisely the requested Pareto-front visualizations:
>   - Identity vs. MFE
>   - scMCC vs. RMSD
>   - Structural vs. energetic cost (and β / round effects)
> - We identify RiboPO Round 2 as lying on a **dominant Pareto frontier** relative to gRNAde and ablated variants, providing concrete evidence of balanced multi-objective optimization.
>
> ---
>
> ### W5 & Q5: Lack of runtime and efficiency analysis.
>
> **Concern.** As a practical fine-tuning framework, RiboPO should report runtime and sample efficiency.
>
> Our improvements (Appendix A.1):
>
> 1. **Inference latency.**
>    On an RTX A6000 (length ≈67, batch size 8), RiboPO uses the same decoder as gRNAde and requires **≈0.66 s per generated sequence**, comparable to diffusion-based baselines with 50–100 denoising steps. Autoregressive generation scales linearly in sequence length and does not require tuning the number of diffusion steps.
>
> 2. **Offline preference construction cost.**
>    Generating ~14k preference pairs (RhoFold+ forward passes) takes about **3 days on a 32-core CPU**. This is a **one-time offline cost** that can be amortized across experiments.
>
> 3. **Training time and convergence.**
>    One DPO round (10 epochs) takes about **16 hours on a single A100 GPU**, with validation metrics saturating around **8k steps**, indicating good sample efficiency for offline RL.
>
> 4. **Sample efficiency / total time to success.**
>    Using the **pass@k** metric, we show that RiboPO achieves an **≈11% higher pass@64** (and larger gains at small k) than gRNAde. This means that **fewer sequences need to be generated and folded** to find acceptable designs, reducing end-to-end time in practical workflows.
>
> In **Appendix A.1**, we also include the time complexity of our models.

---

> ### Author Response · Authors · 2025-12-01
>
> ### W6: remarks on fixed reference policy and surrogate bias.
>
> We now explicitly discuss in **Appendix A.9.3–A.9.4** that:
> - The **fixed reference policy** $\pi_\text{ref}$ acts as a stable prior and trust region, preventing catastrophic drift. We compare fixed vs. updated reference and show that the fixed-reference, multi-round scheme provides the best Pareto trade-off. Plus, our **Table 5** also justified the better overall performance of our fixed-reference setting.
>
>
> ### W6 & Q7: On reliance on surrogate predictors (RhoFold+, ViennaRNA) and robustness to other tools (NUPACK, RNAstructure, etc.)
>
> **Concern.** Since the feedback signals come from RhoFold+ and ViennaRNA, the model may overfit to the specific biases of these tools rather than learning transferable physical principles. How robust are the results to replacing them with other predictors (e.g., NUPACK, RNAstructure)?
>
> Our improvements:
>
> 1. **Cross-tool thermodynamic robustness (NUPACK & RNAstructure).**
>    To directly test whether RiboPO is overfitting ViennaRNA, we added a **cross-tool evaluation** in **Appendix A.6**:
>    - We re-evaluate sequences generated by gRNAde and RiboPO (Round 2) using **NUPACK** and **RNAstructure**, which are *not* used during training.
>    - On these independent folding engines, RiboPO still shows **consistent gains in thermodynamic favorability** (≈+10–12% relative improvement in MFE over gRNAde).
>    This indicates that RiboPO has not simply learned to exploit ViennaRNA-specific quirks; instead, it produces sequences that are more stable **across different energy models**.
>
> 2. **Cross-predictor 3D robustness (AlphaFold3 & DRFold2).**
>    Similarly, to test reliance on **RhoFold+**:
>    - We evaluate gRNAde and RiboPO designs using **AlphaFold3** and **DRFold2**, neither of which is used as a training signal.
>    - Under both predictors, RiboPO designs achieve **lower RMSD (*2.81* for AF3, *2.79* for DRFold2) and higher TM-scores (*0.43* for AF3, *0.39* for DRFold2)** than gRNAde (RMSD: *3.00* for AF3, *2.96* for DRFold2; TM-score: *0.41* for AF3, *0.37* for DRFold2, confirming that the gains in 3D quality are not tied to RhoFold+ alone.
>    This suggests that RiboPO learns structural patterns that generalize to independent 3D models.
>
> 3. **Using multiple surrogates by design.**
>    We emphasize in **Section 3.2–3.4** and **Appendix A.9** that we intentionally use a **vector of surrogates**—RhoFold+ (3D geometry), EternaFold (secondary structure), and ViennaRNA (thermodynamics)—rather than a single predictor. In the theoretical analysis (A.9.2), we model the resulting reward as a **multi-tool surrogate** $r_\Phi$, and argue that:
>    - Only sequences that improve **simultaneously** under several independent tools survive the gating and margin tests,
>    - Which reduces the risk of “reward hacking” any one predictor and encourages learning of more **tool-agnostic physical regularities**.
>
> 4. **Ensemble-level validation (beyond single-structure scores).**
>    In the SSTT benchmark, we also report **ensemble defect (ED/nt)** in addition to single-structure metrics like MFE and RMSD. RiboPO reduces ED/nt relative to gRNAde when evaluated with alternative engines, indicating that it improves **ensemble-level behavior**, not just specific MFE predictions of one tool.
>
> While RiboPO is trained using RhoFold+ and ViennaRNA, the new experiments in Appendix A.6 show that its improvements in both thermostability and 3D structure **persist under NUPACK, RNAstructure, AlphaFold3, and DRFold2**. Combined with the multi-tool reward design and ensemble-level metrics, this provides strong evidence that RiboPO is not overfitting to the idiosyncrasies of any single surrogate, but instead learns **transferable physical principles**.

---

### Official Review · Reviewer_XPtX · 2025-11-01

**Soundness:** 3
**Presentation:** 3
**Contribution:** 3
**Rating:** 8
**Confidence:** 5

**Summary:**

This paper tackles the problem of structural/ensemble property optimization in RNA molecule inverse design. The proposed methodology focusses on Reinforcement Learning and preference optimization algorithms tailored for 3D RNA structure. The experimental evaluation shows the benefits of RL-driven property optimization compared to baselines trained with supervised learning, from both a statistical as well as biological relevance.

**Strengths:**

- The problem being tackled is extremely significant, as RNA inverse design methods are becoming practically used. The paper identifies a key research gap and tackles the problem of RL-driven property optimization in RNA design very well. I particularly want to commend authors’ efforts to develop techniques that are not just copying what is done in the proteins-ML realm, but to do something that’s original and RNA-specific. To the best of my knowledge, this is the first work to tackle this important problem.

- I really enjoyed reading this paper. The exposition does a very good job at presenting technical/methodological ideas and motivating them with ideas from RNA biology, as well as interpreting the results not only from a statistical perspective but also from a biological lens. The use of bolded sentences really makes the manuscript easily graspable immediately.

- The evaluation and experimental setup is rigorous and described in great depth. The ablation studies rigorously analyse various components of the proposed methodology well, and quantify how much each contributes to overall performance.

- I found the results convincing. They support the main claims of the paper well. The analysis of the results goes into sufficient depth about the implications and findings.

- Overall, I believe this is a high quality paper tackling an important problem. I believe that the results and model (if open source) will be of considerable interest to both ML and RNA biology communities, especially as methods like gRNAde and RhoDesign have been validated in wet lab experiments.

**Weaknesses:**

Though not necessarily a weakness of this paper alone, the quality of RNA 3D structure prediction is pretty poor at the moment. Thus, its not surprising that the method does not yet lead to significant gains in terms of 3D metrics over gRNAde, as the structure predictor being used is not reliable. I believe that upcoming 3D structure predictors could push the state of the art further, and then further improve RiboPO as well.

Other than these, I do not see any major weaknesses worth highlighting with this paper. I think its in excellent shape, but I will watch out for other reviewers’ concerns.

**Questions:**

I don’t have major questions.

I have some minor comments:
- I would be interested to see one or two case studies where the RiboPO model improves the ensemble/structural properties of some designs compared to the base model (gRNAde).
- Line 51: Ganser et al citation is for the wrong line, I believe.
- Consider adding some variances/error bars/standard deviations to all of the results reported in tables.

---

> ### Author Response · Authors · 2025-11-27
>
> We are grateful for your very positive assessment and constructive suggestions. Below we address your points one by one.
>
> ---
>
> ### W1: Limited gains in 3D metrics due to current RNA 3D predictors.
>
> **Observation.** Given the current quality of RNA 3D prediction, large gains in 3D metrics are difficult to expect; future predictors may unlock more.
>
> We fully agree, and now:
> - Explicitly acknowledge in Appendix **A.6–A.7** that **RNA 3D prediction is still relatively immature**, limiting the absolute gains measurable in 3D metrics.
> - At the same time, using **AlphaFold3 and DRFold2** as independent evaluators, we show that **RiboPO designs achieve lower RMSD and higher TM-scores than gRNAde**, demonstrating that within current predictor limitations, RiboPO does improve cross-tool 3D structural quality.
>
> ---
>
> ### Q1: Case studies where RiboPO improves ensemble/structural properties vs. the base model.
>
> **Request.** Provide 1–2 qualitative case studies showing how RiboPO improves properties compared to gRNAde.
>
> Our improvements:
>
> We added **Appendix A.10 – Visual Case Studies** with two concrete examples:
>
> 1. **Secondary-structure topology (example: 4MEG_1\_B).**
>    - A forna-based 2D visualization shows that **gRNAde** produces a hallucinated multi-way junction that does not exist in the native fold.
>    - **RiboPO** recovers the correct architecture, separating the 5′/3′ domains appropriately, illustrating how thermodynamic feedback discourages over-knotted, unstable topologies.
>
> 2. **Tertiary structure overlays (example: 5C7W_1\_C).**
>    - We show **AlphaFold3-based 3D overlays** (native vs. gRNAde vs. RiboPO).
>    - gRNAde exhibits misoriented helices and potential steric clashes, while RiboPO maintains a tighter global superposition with the native backbone, consistent with improved RMSD and TM-score.
>
> These case studies visually support the claim that RiboPO produces **structurally and thermodynamically more plausible designs**.
>
> ---
>
> ### Q2: Ganser et al. citation (Line 51).
>
> **Comment.** The Ganser citation previously appeared on the wrong line.
>
> Thanks so much for pointing that out. The citation to **Ganser et al.** now appears in the correct context in the Introduction, attached to the statement about **RNA’s conformational flexibility and multiplicity of competing structures in solution**.
>
> ---
>
> ### Q3: Variances / error bars for reported results.
>
> **Request.** Provide standard deviations or error bars to demonstrate robustness.
>
> Our improvements:
>
> - We added **Appendix A.8** with **Mean ± Standard Deviation** across **four independent random seeds** (for both gRNAde and RiboPO Round 2) on key SSTT metrics (Rec, scMCC, RMSD, MFE).
> - The results show that **RiboPO’s gains are consistent across seeds**, with small variance and no seed for which gRNAde dominates RiboPO across all objectives.

---

> > ### Comment · Reviewer_XPtX · 2025-11-28
> >
> > Thanks for preparing a comprehensive rebuttal.

---

### Official Review · Reviewer_2HaV · 2025-11-04

**Soundness:** 2
**Presentation:** 2
**Contribution:** 2
**Rating:** 2
**Confidence:** 5

**Summary:**

This paper incorporates structural and thermodynamic criteria into RNA design by applying Direct Preference Optimization (DPO) to fine-tune an existing RNA generative model.

**Strengths:**

1. Introducing multi-objective optimization into RNA design is a meaningful and timely direction.
2. RiboPO demonstrates improved secondary-structure consistency and thermodynamic metrics.

**Weaknesses:**

1. The paper introduces a highly complicated evaluation framework, SSTT (Section 3.4), which consists of 15 different metrics. However, most of these metrics are not actually used in the model optimization process. Preference construction relies only on pLDDT, RMSD, and MFE, and the reported results primarily focus on a small subset of about seven metrics. As a result, the necessity and practical value of introducing such a complex evaluation framework are unclear.
2. The central motivation of the paper is to use DPO to encourage the generation of thermodynamically stable RNA structures. However, the approach to “stability” is based solely on ViennaRNA’s minimum free energy (MFE) prediction. MFE is an inadequate surrogate for true thermodynamic stability, and the predicted MFE structure often does not correspond to the experimentally adopted conformation. Therefore, optimizing MFE does not necessarily imply improved thermodynamic robustness or biological stability of the designed RNAs.
3. The MFE structure is typically not the experimentally determined structure present in the dataset, nor is it the structure predicted by data-driven models such as RhoFold or EternaFold. The paper combines predictions from mutually inconsistent tools: RhoFold is used to predict tertiary structure, EternaFold is used to predict secondary structure, and ViennaRNA is used to derive the MFE structure for energy estimation. These three tools produce different structures, meaning that the optimization process is guided by multiple, incompatible structural targets. Using these conflicting structural definitions simultaneously raises concerns about the biological validity of the optimization objective.
4. The paper does not explicitly address the trade-off between multiple objectives. Although the preference construction incorporates both structural fidelity and energy, the relative priority between these two objectives is never clarified. It remains unclear how the method balances potentially conflicting goals, or how the optimization procedure avoids collapsing toward one objective at the expense of the other. Without a principled mechanism for multi-objective trade-off, it is unlikely that the method can reliably achieve a desirable balance.
5. In the ablation study, removing the SFT loss improves the energy objective while degrading structural metrics. This observation does not necessarily justify the inclusion of the SFT term; rather, it simply indicates that the model has moved to a different point on the Pareto frontier, possibly by chance. This further highlights the need for an explicit treatment of multi-objective trade-offs, which is currently lacking in the paper.
6. The paper states that “In Section 4.4.2 we show that these shifts correspond to movement into more designable basins” (line 112). However, the term “designable” is not clearly defined, nor is the notion of a “designable basin” formally introduced or supported. Moreover, I was unable to locate a Section 4.4.2 in the paper.
7. In Figure 1, the filter condition includes an MFE criterion, but this term does not appear in Equation (3).

**Questions:**

See the weaknesses part.

---

> ### Author Response · Authors · 2025-11-27
>
> We thank the reviewer for their careful and technically detailed critique. We have substantially revised the manuscript to address all points, including simplifying SSTT, adding Pareto-front and theoretical analysis, and clarifying the use of surrogate tools and MFE.
>
> ---
>
> ### W1: SSTT is overly complex (15 metrics) while only a few are used in optimization; unclear necessity and value.
>
> **Concern.** The SSTT framework appears excessively complicated relative to the actual optimization signals (pLDDT, RMSD, MFE), and it was unclear why such a broad suite is needed.
>
> **What we changed.**
>
> 1. **Trimmed and clarified SSTT.**
>    Section **3.4** now presents SSTT as a **4-axis framework** (Sequence, Secondary, Tertiary, Thermostability) with a *minimal, non-redundant* set of metrics. Highly correlated or redundant quantities (e.g., melting temperature) were removed or moved to the appendix. The main text now focuses on:
>    - **Sequence:** recovery + k-mer diversity
>    - **Secondary:** EternaFold scMCC
>    - **Tertiary:** TM-score / GDT / RMSD / pLDDT + clash score + INF
>    - **Thermostability:** MFE + ensemble defect (ED/nt)
>
> 2. **Explicit role: optimization vs. evaluation.**
>    We now clearly state that **SSTT is an evaluation/diagnostic suite, not a reward function**. The optimization uses three robust proxies (pLDDT, RMSD, MFE), while SSTT checks whether optimizing these proxies improves broader structural and ensemble properties and does not introduce regressions on unmonitored metrics. This separation is made explicit in Section **4.2** and Appendix **A.2–A.3**.
>
> 3. **Evidence that the “small subset” reward improves the full SSTT suite.**
>    Appendix **A.3** reports full SSTT results (including INF, clash score, ED/nt). Optimizing for the “small subset” of MFE/RMSD/pLDDT **also improves interaction network fidelity and clash scores and reduces ensemble defect (ED/nt)** relative to gRNAde and other baselines, confirming that these proxies are representative of overall physical quality.
>
> The revised SSTT eval results shows that optimizing a small, carefully chosen set of physical proxies improves a broad range of downstream structural and ensemble metrics.
>
>
> ### W2: MFE is an inadequate surrogate for biophysical stability; optimizing ViennaRNA MFE may not yield true thermodynamic robustness.
>
> **Concern.** MFE is an imperfect surrogate; relying solely on ViennaRNA’s MFE may not produce truly robust RNAs.
>
> **What we changed.**
>
> 1. **Explicitly acknowledging limitations of MFE.**
>    In the **Appendix** we now explicitly emphasize that **MFE is a proxy** and can be Goodharted if used in isolation. Our goal is not to claim that ViennaRNA MFE perfectly predicts biological stability, but to show that **RLPF can improve standard thermodynamic proxies in a controlled way**.
>
> 2. **Cross-tool thermodynamic robustness.**
>    To test whether we overfit ViennaRNA, we added a **cross-tool evaluation (Appendix A.6)**:
>    - We evaluate RiboPO and gRNAde using **NUPACK** and **RNAstructure**, neither of which is used during training.
>    - On these independent engines, **RiboPO Round 2** improves MFE by about **+10.2% (NUPACK)** and **+12.3% (RNAstructure)** over gRNAde.
>
>    This shows that the learned policy improves **general thermodynamic favorability**, not just one tool’s MFE.
>
> 3. **Ensemble-level metrics beyond single-structure MFE.**
>    Within SSTT’s Thermostability axis we now also report **ensemble defect (ED/nt)**. RiboPO reduces ED/nt compared to baselines, indicating that **the entire Boltzmann ensemble is shifted toward the target structure**, not just the single MFE structure.
>
> 4. **Theory: surrogate rewards and noisy preferences.**
>    Appendix **A.9.2–A.9.3** models our surrogate reward $r_\Phi$ (constructed from MFE + structural metrics) as a noisy approximation to a latent physical reward $r^*$. Under mild assumptions on the mis-ranking probability, the DPO objective is shown to optimize a **KL-regularized version of $r^*$**. Using a vector of surrogates (3D, 2D, thermo) rather than a single MFE term reduces the likelihood of overfitting to any one tool.
>
> To sum up, we agree MFE is imperfect but standard. The new cross-tool results and ensemble-defect improvements show that RiboPO improves thermodynamic behavior **across independent engines and at the ensemble level**.

---

> ### Author Response · Authors · 2025-12-01
>
> ### W3: Using mutually inconsistent tools (RhoFold, EternaFold, ViennaRNA) may yield conflicting structural targets and undermine biological validity.
>
> **Concern.** The model is guided by multiple, potentially incompatible predictors. How can this produce biologically meaningful optimization?
>
> Our improvements:
>
> 1. **Clarified design rationale in the main text.**
>    In Sections **3.2–3.4**, we explicitly frame the reward as coming from a **multi-tool feature vector**:
>    - Tertiary geometry via RhoFold+ (RMSD, pLDDT)
>    - Secondary structure via EternaFold scMCC
>    - Thermodynamics via ViennaRNA MFE and ensemble metrics
>
>    The goal is **not** to enforce exact agreement between tools but to require that winners be **consistently better across several independent, biased views of the same physical object**.
>
> 2. **Theory: why multi-tool feedback helps.**
>    Appendix **A.9.2** analyzes the case where we use a **surrogate feature vector** $\Phi(s,G)$ built from multiple tools. Using several tools makes it harder to “reward hack” any single predictor; only sequences that look better to *multiple* tools survive feasibility gating and margin checks, yielding a more faithful approximation to the true physical reward.
>
> 3. **Independent structural validation under different 3D predictors.**
>    We added **independent 3D evaluation with AlphaFold3 and DRFold2**, which are not used during training. On these predictors, **RiboPO designs achieve lower RMSD and higher TM-score than gRNAde**, confirming that the improvements are not tied to RhoFold+ alone.
>
> To sum up, multiple predictors are used **by design** to approximate a more robust physical reward. Cross-tool gains in NUPACK/RNAstructure and AlphaFold3/DRFold2 demonstrate that the learned policy generalizes beyond any single surrogate.
>
> ---
>
> ### W4 & W5: No explicit multi-objective trade-off; SFT ablation may simply move to a different Pareto point; missing principled Pareto analysis.
>
> **Concern.** It is unclear how structural vs. thermodynamic objectives are balanced, and whether the SFT term actually improves the trade-off rather than just moving along a Pareto front.
>
> Our improvements:
>
> 1. **Explicit Pareto-front analysis (Appendix A.5, Fig. 3).**
>    We added Pareto plots showing trade-offs between key objectives:
>    - **Fig. 3A: Sequence Identity vs. MFE.** RiboPO Round 2 trades a **∼3% drop in recovery** for a **12.3% improvement in MFE**, showing a controlled movement toward more stable, slightly non-native sequences rather than uncontrolled collapse.
>    - **Fig. 3B: scMCC vs. RMSD (“Designability Frontier”).** RiboPO Round 2 moves the operating point toward **higher scMCC and lower RMSD** relative to gRNAde and later rounds, defining an improved Pareto front.
>    - **Fig. 3C: structural vs. energetic cost.** Different β values and rounds occupy different regions along this Pareto frontier; Round 2 again appears as the best compromise.
>
> 2. **Analyzing the role of SFT and β.**
>    Section **4.4** and Appendix **A.9.4** clarify:
>    - Removing SFT (i.e., $λ_{SFT} = 0$) yields more negative MFE but **substantially worse 3D metrics**, indicating collapse into low-energy but geometrically implausible sequences.
>    - Sweeping β shows that small β (strong KL to the reference) preserves structure but yields smaller MFE gains, while large β improves MFE at a cost to structural fidelity. The chosen β is explicitly motivated as a **Pareto-optimal compromise**.
>
> 3. **Theory: DPO as a KL-constrained multi-objective optimizer.**
>    Appendix **A.9.1–A.9.4** shows that our DPO objective is equivalent to optimizing a **KL-regularized reward**
>    $$
>      r(s,G) = w^\top \phi(s,G),
>    $$
>    where $\phi(s,G)$ contains structural and thermodynamic features, and that the resulting optimal policy
>    $$
>      q^*(s|G) \propto \pi_\text{ref}(s|G)\exp(\beta r(s,G))
>    $$
>    is the solution of this multi-objective trade-off. This ties the observed Pareto fronts directly to the learning objective.
>
> To sum up, the revised manuscript now **explicitly frames RiboPO as a multi-objective optimizer**, provides **Pareto-front visualizations**, and explains how SFT and β shape the trade-off instead of merely moving arbitrarily along it.

---

> ### Author Response · Authors · 2025-12-01
>
> ### W6: “Designable basins” and “designability” not defined; missing Section 4.4.2.
>
> **Concern.** “Designable basins” was previously informal and referred to a non-existent section.
>
> Our improvements:
>
> 1. **Definition in the Introduction.**
>    We now define **Designability** explicitly in the Introduction as the **density or number of sequences that fold into a given target structure**, citing classic work on designability. We also define **designable basins** as regions in sequence space with high density of such sequences.
>
> 2. **Formal definition in the theory section (Appendix A.9.3).**
>    We introduce the **designable set**
>    $$
>    S_{\text{des}}(G) = \{s : \text{RMSD} \leq \tau_{\text{struct}},\ \text{scMCC} \geq \tau_{\text{struct}},\ \text{MFE} \leq \tau_{\text{thermo}}\},
>    $$
>    and define the **designability under a policy** $\pi$ as
>    $$
>    D_\pi(G) = \sum_{s \in S_{\text{des}}(G)} \pi(s|G).
>    $$
>    We then interpret DPO updates as moving probability mass into $S_{\text{des}}(G)$, i.e., into more designable basins.
>
> 3. **Corrected pointer.**
>    The text that previously referred to a non-existent “Section 4.4.2” now correctly points to the **Pareto analysis in Appendix A.5**, which empirically supports the designability picture.
>
> **Summary.** Designability and designable basins are now **precisely defined and linked to the DPO updates**, with the Pareto analysis providing empirical evidence.
>
> ---
>
> ### W7: In Figure 1, the filter condition includes an MFE criterion, but MFE does not appear in Eq. (3).
>
> **Concern.** Inconsistency between the schematic (Figure 1) and the formal equations.
>
> We're so sorry for the confusion. There is a mistake in the filter condition of the previous Figure 1. Our Improvements:
>
> We have **corrected Figure 1** so that:
> - The **Filter condition** matches Eq. (3) and uses only RMSD < γ and pLDDT > κ.
> - The **Margin condition** block (corresponding to Eq. (4)) combines **structural margins (pLDDT, RMSD)** with the **thermodynamic preference** $\text{MFE}(s_w) < \text{MFE}(s_l) $.
>
> The updated figure now exactly reflects Eqs. (3–4) and the implemented pipeline; MFE is not used as a hard filter.

---

### Author Response · Authors · 2025-11-28

We sincerely thank the reviewers (**2HaV, XPtX, sHsX, E3Lp**) for their thoughtful and rigorous feedback.
We have substantially revised the manuscript (changes marked in **red**) with **new experiments, additional theory, and clearer exposition**.

Below we summarize, in a consolidated way, how we addressed the main dimensions of concern.

---

| **Dimension** | **Reviewer 2HaV** | **Reviewer XPtX** | **Reviewer sHsX** | **Reviewer E3Lp** | **Action / Summary of Rebuttal** |
| --- | --- | --- | --- | --- | --- |
| **1. Data Integrity & Rigor** | – | “Add variances / error bars for reported results.” | – | “Explicit overlap between test set and training data of baselines compromises validity.” | **Strict de-duplication (critical fix).** We removed DAS testing dataset of any target appearing in the training data of RiboDiffusion / RDesign and related baselines. After removing the overlapped data for every those baseline models, their performance degrades substantially (e.g., RiboDiffusion RMSD ≈3.7 Å → ≈13.5 Å), while **RiboPO Round 2 remains state-of-the-art** on combined structural + thermodynamic metrics (Appendix A.4). **Variance & robustness.** We re-ran gRNAde and RiboPO (Round 2) across **4 random seeds** and report **Mean ± Std** for key SSTT metrics in Appendix A.8. RiboPO’s gains (especially in MFE and scMCC) are consistent with low variance, supporting statistical robustness. |
| **2. Benchmark Design & Metrics (SSTT)** | “SSTT has 15 metrics, most not used in optimization; practical value unclear.” | – | – | – | **SSTT simplification and clarification.** Section 3.4 now presents SSTT as a **4-axis benchmark** (Sequence, Secondary, Tertiary, Thermostability) with a **reduced, non-redundant set** of metrics. Highly correlated measures (e.g., melt T) were moved to the appendix. **Evaluation vs reward.** We explicitly clarify that **SSTT is a diagnostic evaluation suite**, *not* the training signal. The optimization uses a small, robust subset (RMSD, pLDDT, MFE), and SSTT is used to check for unintended regressions and to validate that improvements generalize to other axes (INF, clash, ensemble defect). Full SSTT results (Appendix A.3) show that optimizing this subset **improves interaction networks, clash scores, and ensemble defect**, confirming its practical value. |
| **3. Biophysical Validity & Surrogates** | “MFE is an inadequate surrogate; inconsistent use of RhoFold, EternaFold, ViennaRNA may harm biological validity.” | – | “Reliance on surrogate predictors (RhoFold+, ViennaRNA)… overfitting rather than learning physical principles?” | – | **Cross-tool thermodynamic robustness.** To test reliance on ViennaRNA, we evaluate gRNAde and RiboPO using **NUPACK** and **RNAstructure**, which were *not* used in training. RiboPO Round 2 achieves **≈10–12% better MFE** than gRNAde under these engines (Appendix A.6), showing that stability gains **generalize beyond a single tool**. **Cross-predictor 3D robustness.** Similarly, we evaluate designs with **AlphaFold3** and **DRFold2** instead of RhoFold+. RiboPO designs obtain **lower RMSD and higher TM-scores** than gRNAde, showing that 3D improvements are not tied to RhoFold+ alone. **Multi-surrogate design.** We now emphasize that our reward uses a **vector of surrogates** (RhoFold+ for 3D, EternaFold for 2D, ViennaRNA for thermo). In Appendix A.9, we model the resulting reward as a **multi-tool surrogate** $r_\Phi$ and argue that requiring improvements across several independent tools reduces “reward hacking” of any one predictor. **MFE role clarified.** We clarify that MFE is used as a **margin criterion** in preference construction (not a hard filter), and Figure 1 has been corrected so its “filter” and “margin” blocks now match Equations (3) & (4) exactly. |

---

### Author Response · Authors · 2025-12-01

(Cont.)

| **Dimension** | **Reviewer 2HaV** | **Reviewer XPtX** | **Reviewer sHsX** | **Reviewer E3Lp** | **Action / Summary of Rebuttal** |
| --- | --- | --- | --- | --- | --- |
| **4. Trade-offs, Multi-objective Behavior & Designability** | “Trade-off between structural fidelity and energy is never clarified; ‘designability’ not defined.” | – | “Decline in sequence recovery; missing Pareto analysis; unclear if multi-objective optimization... trading one metric for another.” | – | **Explicit Pareto-front analysis.** Appendix A.5 now includes **Pareto plots**: (i) Sequence Identity vs. MFE, (ii) scMCC vs. RMSD (“Designability Frontier”), and (iii) structural vs. energetic cost, across β values and DPO rounds. These show that **RiboPO Round 2** lies on a **dominant Pareto frontier**: a small (≈3%) drop in identity yields a **large (~12.3%) gain in MFE** and improved scMCC/RMSD compared to gRNAde. **Designability formally defined.** We now define **designability** in the Introduction (density of sequences folding to a target structure) and, in Appendix A.9.3, formalize **designable sets** $S_{\text{des}}(G)$ and **designability under a policy** $D_\pi(G)$. We show conceptually that DPO updates move probability mass into $S_{\text{des}}(G)$, i.e., into more designable basins. **Controlled trade-off, not collapse.** The new analysis makes clear that the observed drop in sequence recovery is a **controlled trade-off** that moves the model to **more stable, structurally consistent regions**, rather than an unprincipled regression. |
| **5. Efficiency & Practical Applicability** | – | – | “Lack of time and efficiency analysis; practical cost vs baselines unclear.” | – | **Runtime and sample-efficiency analysis (Appendix A.1).** We now report: (i) **Inference latency**: ≈0.66 s/sequence on RTX A6000, comparable to diffusion baselines with 50–100 denoising steps; (ii) **Offline preference construction cost**: ≈3 days CPU for ~14k preference pairs (one-time, amortizable); (iii) **Training cost**: ≈16 hours per DPO round on a single A100. **Total time to success.** Using **pass@k** (Table 2), RiboPO achieves ≈11% higher pass@64 than gRNAde, meaning fewer sequences must be generated and folded to obtain valid designs—reducing real-world design time despite the offline RL cost. We also included big-O time complexity analysis at **Appendix A.1**.|
| **6. Novelty & Contribution** | “Introducing multi-objective optimization into RNA design is meaningful and timely.” | “Original and RNA-specific; not just copying protein ML.” | “Reformulating RNA inverse folding as multi-objective preference optimization is conceptually elegant.” | “Framework is mostly application; lacks theoretical justification; contribution may be incremental.” | **Clarified contribution framing.** We revised the Introduction and Discussion to **de-emphasize algorithmic novelty of DPO itself** and to highlight instead: (i) an **RNA-specific RLPF instantiation** that explicitly optimizes **thermodynamic and ensemble stability** (a missing piece in prior RNA design); (ii) the **SSTT benchmark**, the first unified 4-axis evaluation (Sequence, Secondary, Tertiary, Thermostability) for RNA inverse folding; and (iii) a **combined empirical + theoretical analysis of fixed-reference, multi-round DPO** in the RNA energy landscape. **Positioning.** We make clear that many reviewers already view the direction as meaningful and RNA-specific; we add theory (Appendix A.9) and clean benchmarks (Appendix A.4, A.6) to address the concern about “mere application” and to show that the framework is both **principled and practically impactful**. |
| **7. Visual & Qualitative Evidence** | – | “Would like to see case studies illustrating improved ensemble/structural properties.” | “Lack of visual RNA design analysis (2D / 3D overlays).” | – | **New visual case studies (Appendix A.10).** We added qualitative examples where we compare **Native vs. gRNAde vs. RiboPO**: (1) **2D secondary structure (forna)**: gRNAde hallucinates incorrect junctions/topologies, while RiboPO recovers the correct native architecture, illustrating improved secondary-structure realism. (2) **3D overlays (AlphaFold3)**: RiboPO’s designs track the native backbone more closely and reduce apparent steric clashes compared to gRNAde. These visuals concretely demonstrate that RiboPO’s improvements are not only numerical but also **structurally interpretable**. |

---

### Author Response · Authors · 2025-12-01

(Cont.)

| **Dimension** | **Reviewer 2HaV** | **Reviewer XPtX** | **Reviewer sHsX** | **Reviewer E3Lp** | **Action / Summary of Rebuttal** |
| --- | --- | --- | --- | --- | --- |
| **8. Theoretical Foundations** | – | – | – | “No theoretical justification for applying DPO to physical criteria; no analysis of margin/curriculum behavior; conclusions are unprincipled.” | **Theoretical analysis of RLPF with DPO (Appendix A.9).** We now provide a **formal treatment**: (1) We define a latent **multi-objective physical reward** $r(s,G) = w^\top \phi(s,G)$ (combining structural and thermodynamic features) and derive the associated KL-regularized target distribution $q^*(s \mid G) \propto \pi_{\text{ref}}(s \mid G)\exp\big(\beta\, r(s,G)\big)$. (2) We show how our **feasibility gating + margin-based preferences** yield pairwise comparisons consistent with $r$, and under standard noise assumptions, the **DPO gradient is aligned with $\nabla r$**, so DPO moves toward $q^*$. (3) We analyze **multi-round, fixed-reference DPO** as approximate **mirror-descent steps**, with the margin curriculum providing coarse-to-fine preference signals. (4) We discuss **Goodhart-type effects** and show how SFT and the KL term regularize against over-optimization of mis-specified surrogates. (5) Together, this provides a **principled justification** for applying DPO to physical surrogate objectives and explains the observed behavior of $\beta$, margins, and multiple rounds. |

---

In summary, we believe the revised manuscript:

- Uses **cleaner, leakage-free evaluation** with variance analysis,
- Demonstrates **cross-tool, cross-predictor robustness** of both stability and structure,
- Provides **explicit Pareto-front and designability analyses** clarifying trade-offs,
- Includes **practical runtime and pass@k efficiency results**,
- Adds **visual case studies** for interpretability, and
- Supplies a **solid theoretical foundation** for RLPF with DPO on physical surrogates in RNA design.

We hope these substantial revisions address the reviewers’ concerns and clarify both the **novelty** and the **reliability** of our contributions.

---

### Meta-Review · Area_Chair_fAbo · 2026-01-09

**Summary:**

The reviewers raised a number of concerns that in total are pretty well and fairly summarized by the authors. For the purposes of this meta review I think there were some comments that are potentially addressed but required some additions to the paper:

- Lack of error bars in some results (resolved, easily verified as resolved)
- Original evaluation framework was quite complicated. Fair criticism of the original. *Potentially* resolved by the reviewers, but my read of this is as a fairly significant change.
- Reliance on surrogate predictors. Some amount of robustness demonstrated here by verifying with additional surrogates. That's at least somewhat convincing, although there's possibly a question of how correlated the errors across surrogates might be.

**Reviewer Concerns:**

The authors clearly made significant efforts to address *all* points raised by the rebuttal. To the authors' credits, none of their actions appear to have been "cop outs" of attempting to explain away the problems: almost all concerns were met with changes to the papers, methodology, or new experimental results and ablations.

**Reviewer Scores:**

This is really difficult for this paper without reviewer involvement because (1) the reviewers started uniformly from fairly negative starting positions, indicating fairly negative starting positions, but (2) a fairly substantial amount has changed in response to the reviews. I think it's likely there would have been some upward score movement, which leaves the paper as borderline. The problem is that, because *so much* changed in response to reviewer comments, it's hard to comfortably substitute my own judgement here: this wouldn't be simply ignoring some weak reviewer concerns and listening to some positive reviewers, it would be one-person re-reviewing a paper changed in a substantial way.

---

### Decision · Program_Chairs · 2026-01-26

Reject